

# Planckian dissipation, minimal viscosity
# and the transport in cuprate strange metals

**Jan Zaanen**

The Institute Lorentz for Theoretical Physics, Leiden University, Leiden, The Netherlands

jan@lorentz.leidenuniv.nl

## Abstract

Could it be that the matter formed from the electrons in high Tc superconductors is of a radically new kind that may be called "many body entangled compressible quantum matter"? Much of this text is intended as an easy to read tutorial, explaining recent theoretical advances that have been unfolding at the cross roads of condensed matter- and string theory, black hole physics as well as quantum information theory. These developments suggest that the physics of such matter may be governed by surprisingly simple principles. My real objective is to present an experimental strategy to test critically whether these principles are actually at work, revolving around the famous linear resistivity characterizing the strange metal phase. The theory suggests a very simple explanation of this "unreasonably simple" behavior that is actually directly linked to remarkable results from the study of the quark gluon plasma formed at the heavy ion colliders: the "fast hydrodynamization" and the "minimal viscosity". This leads to high quality predictions for experiment: the momentum relaxation rate governing the resistivity relates directly to the electronic entropy, while at low temperatures the electron fluid should become unviscous to a degree that turbulent flows can develop even on the nanometre scale.

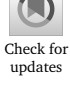
# 1  Introduction

This is not a run of the mill physics research paper. Instead it is intended to arouse the interest of especially the condensed matter experimental community regarding a research opportunity having a potential to change the face of fundamental physics.

The empirical subject is a famous mystery: the linear in temperature resistivity measured in the strange metal phase of the cuprate high Tc superconductors [1]. Resting on the latest developments at the frontiers of theoretical physics, the suspicion has been growing in recent years that this may be an expression of a new, truly fundamental physics. This same paradigm may be of consequence as well for subjects as diverse as the nature of quantum gravity, phenomena observed in the quark gluon plasma and the design of benchmarks for the quantum computer. I will present here the case that in a relatively effortless way condensed matter experimentation may make a big difference in advancing this frontier of truly fundamental physics.

It is metaphorically not unlike Eddington investing much energy to test Einstein's prediction of the bending of the light by the sun. I have some razor sharp smoking gun predictions in the offering (section 6) that should be falsified or confirmed in the laboratory. As in Eddington's case, at least for part of it no new experimental machinery has to be developed. The theory is unusually powerful, revealing relations between physical properties which are unexpected to such a degree in the established paradigm that nobody ever got the idea to look at it. However, it does require a collective effort and the intention of this "manifesto" is to lure the reader into starting some serious investments.

The basic idea has been lying on the shelf already for a while. I did not take it myself too serious until out of the blue data started to appear that are in an eerie way suggestive. There is more but the tipping point was reached when I learned about the results of Legros et al. [2]. These authors demonstrate a remarkably tight relationship between the way that the linear resisitivity and the *entropy* depend on doping in the overdoped regime. My first request to the community is to study specifically the scaling of the *Drude width* (momentum relaxation time) with the entropy, systematically as function of doping and temperature, employing high precision measurements of both the thermodynamics and the (optical) conductivity (see section 6.1).

What is the big deal? The predicted relation between thermodynamics and transport is alien to any type of established transport theory. However, if confirmed it would demonstrate

that the principle behind a famous discovery in the study of the quark-gluon plasma [3] is also at work in the cuprate electron systems. This is the "minimal viscosity" found in the title of this paper (section 4.3). The special significance is that this was predicted [4,5] well before the experiments were carried out at the heavy ion colliders, employing mathematical machinery discovered by the string theorists in their quest to unravel quantum gravity. This is the AdS-CFT correspondence [6] (also called 'holographic duality") that maps the physics of a strongly interacting quantum system onto a "dual" gravitational (general relativity) problem. In this language the minimal viscosity is rooted in universal features of quantum black hole physics [4] – the Hawking temperature affair.

Starting some ten years ago, the "correspondence" was also unleashed on the typical circumstances characterizing the electronic quantum systems in solids. Remarkably, upon translating the fancy black holes associated with these conditions to the quantum side a physical landscape emerges that is surprisingly similar to what is observed experimentally in the high Tc superconductors [7,8]. However, the underlying physics is completely detached from the Fermi-liquids and BCS superconductors of the condensed matter textbooks. This is quite suggestive but it has been waiting for experiments that critically test whether this holographic description has anything to do with real electrons in solids.

A number of years ago we stumbled on the present minimal viscosity explanation for the linear resistivity [9], playing with the black holes. I did not take it seriously until the recent experimental developments. If confirmed, it would represent impeccable evidence for holographic principle to be at work in condensed matter. On the one hand, it rests on truly universal aspects of the physics in the gravitational dual. Perhaps even more significant, *it is extremely simple* at least for the initiates. To explain the significance of this simplicity, I will review the intellectual history of the linear resistivity conundrum in the next section 2 , highlighting Laughlin's proposition that the linear resistivity can only be explained by a simple theory. This section may be of some interest to the holographists since they may not be completely aware of the good thinking during the haydays of the strange metals in the 1990's. Whatever, the case in point is that the holography experts may now jump directly to section 5 to get reminded of the idea, to then convince themselves of the predictions in section 6 in a matter of minutes. It boils down to elementary dimensional analysis.

The qualitative nature of the physics at work is however stunningly different from anything in the condensed matter textbooks. At first encounter, the claims may sound outrageous. The electron fluid in the strange metal is claimed to be governed by hydrodynamics, it flows like water. The difference with water is that the viscosity of the fluid is governed by the strange rule that its magnitude is set by $\hbar$, while it is proportional to the entropy. The entropy in the cuprate strange metal is known to exhibit a peculiar doping dependence and this implies quite a surprise for the doping dependence of the Drude width in the underdoped regime as I will discuss in section 6.1.

The spectacular weirdness of this fluid gets really in focus in section 6.2. The implication is that the viscosity becomes so tremendously small at low temperatures that this fluid will be susceptible to *turbulent flow phenomena on the nanometer scale*. You may check it with any fluid mechanician who will insist that this is impossible. Observing this nano-scale turbulence would proof the case once and forever. However, it is for material reasons quite difficult to construct transport devices capable of measuring this turbulence. When the hypothesis survives the numerology tests of section 6.1, the case may be credible enough for the community to engage in more of a big physics mode of operation aimed at constructing the nano transport devices. Compared to the bills of high energy- and astrophysics it will be remarkably cheap, while the odds are that an army of string theorists will cheer it more than anything that can be anticipated to happen on the short term in these traditional areas of fundamental physics.

This is the original content of this paper. But three other sections are included as well.

These are intended as a concise tutorial, updating the readers who are interested in the physics behind the strange predictions that I just announced. In the last few years there has been actually quite some progress in comprehending this affair. Some five years ago the response to the question of why these things happen would have been an unsatisfactory "because the black holes of the string theorists insist on it, and it looks a bit like experiment". The big difference since then is in the cross-fertilization with quantum information. The string theorists took this up for their own quantum gravity reasons (to get an impression, see ref. [10]). As a beneficial spin-off, it is now much more clear what kind of physics is addressed in the condensed matter systems.

When physics is truly understood it becomes easy. I actually wrote this tutorial to find out whether I could catch this story on paper in simple words. It is not at all intended to be an ego document. To the contrary, from the ease of the conversations with other holographists it appears that we all share the same basic outlook. This may be just a first attempt to formulate this collective understanding in the simplest possible words.

All what is required is some elementary notions of quantum information theory. This involves not much more than recalling basics of quantum many body theory, to then ask the question of how a computer would be dealing with it [11]. The crucial ingredient is *entanglement,* as the property that divides the classical from the quantum. As I will explain in section 3.1, nearly all the physics you learned in condensed matter (and high energy) courses departs from vacuum states that are in fact very special: these are not entangled on a macroscopic scale and they are just classical stuffs described in a wave function language. This includes surely the conventional "quantum liquids": the superconductors and the Fermi-liquid. But different forms of matter may exist and I will explain why the circumstances in the cuprates may be optimal for the formation of such stuff. I like to call it "irreducibly many body entangled compressible quantum matter", or "compressible quantum matter" in short.

It is crucial to be aware of our ignorance: a quantum computer is needed in principle [11] to find out how such stuff works. Now it comes: we have come to the realization that holography is a mathematical machine that computes observable properties of stuff that is some kind of most extreme, "maximally" entangled form of this compressible quantum matter. Its observable properties do represent "physical" physics. However, this can be very different from anything that you learned in college. I will dwell on this further in section 3: first I will explain why we like to call it "unparticle physics" (section 3.2) to end this section with crucial insights of quantum thermalization. The key word is here "eigenstate thermalization" [12]; the equilibration processes work in densely entangled matter in a manner that appears as completely unreasonable trying to comprehend it form inside the "particle physics tunnel" (see e.g. [13,14]). Paradoxically, at the end of the day these may turn into exceedingly simple physics. This is the secret behind the Planckian dissipation [15], the minimal viscosity [5] and even the very fact that a hydrodynamical description may make sense [14].

I will then turn to holography in section 4. This is no more than a tourist guide of the holographic theme park [7]. After introducing the generalities (section 4.1), I will zoom in on the phenomenology of the holographic strange metals (section 4.2). For the present purposes, the most important part is section 4.3 where I will highlight holographic transport theory, spiced up with quantum thermalization principle, where the minimal viscosity is the top attraction.

Equipped with this background information it should become crystal clear what is going on in the "technical" sections 5 and 6. I do hope that with these added insights the reader will share my sentiment that the venture proposed in these chapters is perhaps the most exciting research adventure that can be presently imagined in all of fundamental physics.

## 2  Why is the linear resistivity an important mystery ?

It is observed that the resistivity as function of temperature is a perfect *straight line* in the strange metal regime of the normal state of high Tc superconductors [17], from the superconducting transition temperature up to the highest temperatures where the resistivity has been measured [18]. This was measured for the first time directly after the discovery of the superconductivity at a high temperature in 1987. Myriads of theories were forwarded claiming to explain this simple behavior but more than thirty years later there is still no consensus whether any of these even make sense [1]. Bob Laughlin formulated a long while ago a criterium to assess the quality of linear resistivity theories [19], that to my opinion has stood the test of time. To paraphrase it in my own words: "*the linear resistivity reflects an extremely simple physical behaviour and extremely simple behaviours in physics need a powerful principle for their protection. Explain this principle.*"

Without exception, all existing proposals fail this test. Most of these theories depart from the assumption that the electrical currents are carried by one or the other system of quasiparticles. This is usually a Fermi gas that is coupled to one or the other bath giving rise to a lifetime which in turn is directly related to the transport life time (e.g., [20]). This can give rise to linear resistivities but the problem is that this fails to explain why nothing else is happening. This is fundamental: it is impossible to identify the simplicity principle dealing with *particle physics*. The transport is assumed to be due to thermally excited quasiparticles behaving like classical balls being scattered in various ways, dumping eventually their momentum in the lattice. However, it is a matter of principle that the physics of such quasiparticles in real solids is never simple. These interact with phonons which are very efficient sources of momentum dissipation which should be strongly temperature dependent for elementary reasons. When the inelastic mean free path becomes of order of the lattice constant the resistivity should saturate. In the cuprates there is just nothing happening at the temperature where the magnitude of the resistivity signals the crossover from "good" to "bad" metal behavior [21]. The conclusion is that quasiparticle transport *should* give rise to some interesting function of temperature by principle. Laughlin's criterium is violated by the basic principles from which quasiparticle physics departs.

Is there any truly general principle at work that can be unambiguously distilled from experiment? It appears that at frequencies $\omega \leqslant k_B T/\hbar$, the line shape of the optical conductivity as function of frequency is remarkably well described by a classic Drude form [22]

$$\sigma(\omega) = \frac{\omega_p^2 \tau_K}{1 + i\omega\tau_K}, \tag{1}$$

where $\omega_p = \sqrt{ne^2/m}$ is the plasma frequency ($\omega_p^2$ is the Drude weight) while I call $\tau_K$ the "transport momentum relaxation rate". Perhaps the first occasion that this case was convincingly made is in Fig. 2b of ref. [22]. This revealed a remarkably simple number. According to the data the Drude weight is temperature independent while $\tau_K = A\tau_\hbar$ , where

$$\tau_\hbar = \hbar/(k_B T) \tag{2}$$

while $A \simeq 0.7$ in an optimally doped BISCO superconductor. The DC resistivity $\rho(T) = 1/(\omega_p^2 \tau_K)$ and this reveals that the origin of the linear temperature dependence of the resistivity is in the momentum relaxation rate, Eq. (2). To draw attention to this remarkable observation I introduced the name "Planckian dissipation" for the phenomenon [15] (see also [16]).

This is just a discussion of data, what is the principle? A folklore developed last century in the condensed matter community, jumping to the conclusion that a Drude conductivity signals the Fermi-liquid, while $\tau_K$ parametrizes the quasiparticles scattering time. But a Drude

conductivity is actually much more general. This was already understood in the old transport literature. Inspired by holographic transport theory this was recently thoroughly cleaned up by Hartnoll and coworkers [23] mobilizing the memory matrix formalism [24]. The essence is very simple [7]: Drude does not tell anything directly regarding the nature of the matter that is responsible for the transport. A first requirement for a Drude response is that one is dealing with *finite* density; very rare occasions in condensed matter such as graphene at charge neutrality where the density is effectively zero are exempted. Dealing with finite density stuff in the Galilean continuum, regardless whether it is a crystal [25], the strongly coupled quantum fluids of holography or a simple quantum gas, it will always exhibit an optical conductivity of the Drude form for the special value $\tau_K = \infty$. This is the famous "diamagnetic peak", the delta function at zero frequency that tells that the metal is perfect characterized by an infinite conductivity. The reason is very simple. At finite density the electrical field will accelerate the system of charged objects. It acquires a *total* momentum and in a homogenous and isotropic space this total momentum is conserved. The electrical current is proportional to this momentum and since the latter is conserved the former does not dissipate. Upon breaking the translational symmetry total momentum is no longer conserved and will relax. As long as it is "nearly" conserved (the momentum relaxation time is long compared to microscopic time scales) the outcome is that the delta function peak will broaden in a Drude peak with a width set by the transport momentum relaxation time, the $\tau_K$ in Eq. (1). One observes sharp Drude-like peaks in the optical conductivity of cuprates in the lower temperature "good strange metal" regime. We learn therefore that these metals are in an effective near-momentum conservation regime.

But this sheds light on the nature of Laughlin's protection principle. Dealing with conventional quasiparticle physics the physics leading to momentum relaxation is by principle a complicated affair as I already argued. On the other hand, dimensional analysis has a track record as the first thing to do when dealing with unknown physics and the Planckian dissipation is a vivid example. Planck's constant carries the dimension of energy times time. $k_B T$ has the dimension of energy and their ratio – the Planckian relaxation time $\tau_\hbar$ Eq. (2) defines the most elementary quantity with the dimension of time in a quantum system at finite temperature. We learn directly from experiment that this simplest of all dissipative time scales is responsible for the unreasonable simplicity of the linear resistivity [15]!

With help of string theory and quantum information there has been quite some progress in recent years in understanding the origin of this exceedingly simple behavior. As I will explain next, Planckian dissipation appears to be a highly generic property of *densely many body entangled compressible quantum matter*. This is very exciting: the linear resistivity is just reflecting that we are dealing with a completely new form of matter controlled by a system where literally everything is entangled with everything. Since the "mechanism" appears to be highly generic it is of as much relevance to heavy ion collisions as to black hole physics. But in condensed matter physics it can be rather easily studied in the laboratory.

## 3 Compressible quantum matter: Unparticle physics and quantum thermalization

This is a highly tutorial section. I will discuss some of the most elementary notions associated with the use of quantum information language in many body quantum physics. On this level it is mostly about giving new names to phenomena familiar from condensed matter and/or field theory courses. However, addressing these in terms of mathematical information language is instrumental for getting a clearer view on how quantum physics works. On the one hand, basic notions of mathematical complexity theory such as "non-polynomial hard" turn into a

merciless way to convince oneself that there are things going on that are beyond the capacity of any computation, making us acutely aware of our ignorance of large parts of the physical landscape. On the other hand, it is just very insightful that entanglement is the key aspect dividing the (semi)classical matter of the textbooks from genuine, "NP-hard" quantum matter. This revolves around the nature of the ground state and this will be the subject of section 3.1. In the present context the material in section 3.2 is a bit of a detour but I do have the experience that especially experimentalists are quite grateful when they catch this no-brainer: when the groundstate is many body entangled it is impossible to identify particles in the spectrum. Such "unparticle physics" is easy to identify in spectroscopic information since it gives rise to the familiar observation that spectral functions are "incoherent" (actually, a bad name). Section 3.3 is crucial for those who are not yet informed regarding the way that quantum thermalization works. The key word is "eigenstate thermalization hypothesis" (ETH); it is yet again very easy to understand but it is a bit of a shock when one comprehends it for the first time. This makes possible for phenomena to happen that are completely incomprehensible for the particle physicist's mind, while we have in the mean time very good reasons to assert that this machinery is behind the Planckian dissipation.

## 3.1 Quantum matter: non Fermi liquids and many-body entanglement in the vacuum

The effort to build the quantum computer has at the least had the beneficial psychological influence on physics that we now dare to think about realms of reality that not so long ago were so scary that we collectively looked away from it [26]. It is just a good idea to invoke the branch of mathematics called information theory that was actually discovered in the early days of digital computers [27]. For the present purposes one needs to know only some elementary notions of mathematical complexity theory [28]. One has an algorithmic problem with $N$ "bits of information" - for instance, the travelling salesman that has to visit $N$ customers. One has now to write a code computing the most efficient travel plan. The question is, how does the time $t_N$ it takes to compute the answer scale with $N$? The first possibility is that this time increases polynomially, $t_N \sim N^p$: such a problem can be typically cracked using computers that are available in 2018. However, it may be that this increases exponentially instead, $t_N \sim \exp N$. This is called "nondeterministic polynomial hard" or "NP-hard" where "hard" refers to the absence of a trick to map it onto a polynomial problem. The travelling salesman problem is famously NP hard. NP hard problems are incomputable given the exponential rate by which one has to expand the computational resources. One should be aware that it is a very serious condition: when the computers cannot handle it there is neither an elegant system of mathematical equations that can be solved in closed form either.

The issue is that generic quantum many body problems are NP hard [11,29]. When I was a young postdoc in the late 1980's the biggest Hubbard lattice that could be diagonalized exactly was of order 20 sites. Despite the exponential growth of computational resources, in 2018 this has barely improved since then. Why is that?

Part of the quantum computer affair is that any physical system can be reduced to a system of qu-bits taking values of 0 or 1. Similarly, many qu-bit systems live in a Hilbert space spanned by tensor products. A familiar example is the affair dealing with 2 qubits, with the four dimensional Hilbert space $|0\rangle|0\rangle, |0\rangle|1\rangle, |1\rangle|0\rangle, |1\rangle|1\rangle$. Taking coherent superpositions one can form the maximally entangled Bell pairs like $(|0\rangle|0\rangle + |1\rangle|1\rangle)/\sqrt{2}$ with the tricky part that entanglement is representation independent. For instance, the state $(|0\rangle|0\rangle + |1\rangle|0\rangle + |0\rangle|1\rangle + |1\rangle|1\rangle)/2$ is an unentangled product since it can be written as $|+\rangle|+\rangle$ where $|+\rangle = (|0\rangle + |1\rangle)/\sqrt{2}$. What happens when there are three qubits? In total there are $2^3 = 8$ different configurations of three qubits and the Hilbert space becomes 8 dimensional. Similarly for $N$ qubits the Hilbert space dimension becomes $2^N$. Now it comes: one can assign

$10^{23}$ qubits to 1 gram of matter implying a Hilbert space dimension of $2^{10^{23}}$. This is a gargantuan number, realizing for instance that there are only of order $10^{80}$ baryons in the entire universe.

Energy eigenstates are a priori of the form

$$|\Psi_n\rangle = \sum_{i=0}^{2^N} a_i^n |config, i\rangle, \tag{3}$$

where $|config, i\rangle$ is one of the $2^N$ qubit configurations. "Typical" eigenstates are irreducibly many body entangled in the sense that a number of states that is extensive in the big Hilbert space has a finite (albeit very small $\sim 1/\sqrt{2^N}$) amplitude in the coherent superposition while there is no representation that brings it back to a product form. One recognizes immediately that this is NP hard. This is not computable with classical means, while all of reality is made out of this stuff. Why is it so that we can still write physics textbooks? There are two sides to the answer: (a) ground states are special (this section), and (b) quantum ("eigenstate") thermalization (section 3.3).

Let us first zoom in on the ground states. The "vacuum rules" and all states of matter that one encounters in the textbooks of condensed matter and high energy physics are of a very special "untypical" type. In quantum information these are called "short ranged entangled product states" ("SRE products") [30], which are as follows. Take the appropriate representation and instead of Eq. (3) the ground state wave function has the form

$$|\Psi_0\rangle = A|\Psi^{clas.}\rangle + \sum \hat{a}_i^n |config, i\rangle, \tag{4}$$

where a particular product state $|\Psi^{clas.}\rangle$ "dominates" the wavefunction: $A$ is a number of order 1 while only a very limited number of the amplitudes $\hat{a}_i^n \neq 0$. These can be computed perturbatively "around" $|\Psi^{clas.}\rangle$. For example, consider a solid; the appropriate representation is in the form of real space wave packets localized at positions. Form a regular lattice of such wave packets and occupy them with atoms. We immediately recognize the crystal which is a quite classical state of matter but it of course still decribed by a wavefunction. But since this wavefunction is a product it forgets about entanglement and it therefore represents matter that can be described with the classical theory of elasticity. But this is an exagguration since there are still quantum fluctuations giving rise to e.g. a quantum Debye-Waller factor which is observable when the atoms are sufficiently light. These are restored by the standard, rapidly converging perturbative corrections wiring in the $\hat{a}_i^n$ states in the vacuum. These take care that some of the many body entanglement is restored but only up to a small, still microscopic scale. At distances larger than this "entanglement length" all the physics of this state will become precisely classical since the entanglement that makes the difference has vanished.

The solid is surely a very obvious example of how to make classical stuff in the macroscopic realms from quantum parts. However, the same basic notion applies to *anything* described by conventional Hartree-Fock mean field theory. This includes all forms of states that break symmetry spontaneously, but also the states that were called "quantum fluids" in the past. This is just a matter of single particle representation: these are SRE products in single particle momentum space instead. For the Fermi-liquid $|\Psi_{FL}^{clas.}\rangle = \Pi_k c_k^\dagger |vac.\rangle$; this is of course yet again a pure product. It is "enriched" by the Pauli principle but the case can be made precise [31] that the Pauli principle wires in only classical information. The BCS wavefunction is in turn a textbook example of a product: $|\Psi_{FL}^{clas.}\rangle = \Pi_k (u_k + v_k c_k^\dagger c_{-k}^\dagger)|vac.\rangle$. Such SRE products are just the wavefunction way of encapsulating conventional mean field theory and such vacuum states typically represent the spontaneous breaking of symmetry (e.g., the BCS orderparameter $\sim \sum_k u_k v_k$).

As a caveat, the topologically ordered states of the strongly interacting variety are incompressible, characterized by an energy gap separating the ground state from all excitations. This gap enforces short range entanglement, except that there is room left for some extremely sparse non-product state structure that is then responsible for the spooky, immaterial topological properties [26, 30]. Since the fractional quantum Hall revolution we have quite a good idea of how this works. However, what is known about the quantum nature of ground states of systems forming *compressible* (gapless) matter? The answer is: nearly nothing.

There is surely no such thing as a theorem insisting that all ground states have to be of the product type. In fact, there is one state that is irreducibly many particle entangled that is reasonably well understood: the ground state associated with the strongly interacting, non-integrable quantum critical state realized precisely at the quantum critical point. This is the material familiar from Sachdev's book [32]. This is best understood in Eucliden path integral language: the quantum system maps onto an equivalent *classical* statistical physics problem in one higher dimension where the extra dimension corresponds with imaginary time. At the critical coupling, the Euclidean incarnation corresponds with the classical system being precisely at the critical temperature where the system undergoes a continuous phase transition and a thermal critical state is realized. As is well known in the statistical physics community, when this critical state is strongly interacting (below the upper critical dimension) it is actually NP hard. The polynomial method (Metropolis Monte Carlo) fails eventually right at the critical point because an extensive number of all configurations take part in the classical partition sum. The path integral just represents the vacuum state which is therefore of the form Eq. (3). In this context, one finds a first hint regarding the origin of Planckian dissipation: for simple scaling reasons as explained in Sachdev's book one finds that generically the quantum linear response functions at finite temperatures are characterized by Planckian relaxation times (see section 3.3).

In such "bosonic" systems one has to accomplish infinite fine tuning to reach the quantum critical point and such "bosonic" matter is therefore generically of the SRE product kind. However, the mapping of the quantum system on an equivalent statistical physics problem requires actually quite special conditions: it only works when the ground state wavefunction can be written in a form that *all* amplitudes $a_i^0$ are positive definite. One needs here special symmetry conditions, most generally charge conjugation and time reversal. Otherwise the system is formed from bosons where one can easily prove that the amplitudes are positive definite. Dealing with strongly interacting fermions at a finite density the condition is never automatically fulfilled: this is the infamous fermion sign problem. The trouble is that the quantum system no longer maps on an equivalent stochastic, polynomial-complexity statistical physics problem since it is characterized by "negative probabilities". The claim is that the sign problem is *NP hard* [29]. Although great strides forward has been made due to a resilient effort in the computational community (e.g., [33]), it is generally acknowledged that nothing is known with certainty regarding the long wavelength and low temperature physics of such fermion problems. As I just emphasized, this NP hardness is equivalent to the statement that the vacuum state is irreducibly many body entangled in the sense of Eq. (3). The implication is that when the system does not renormalize in a Fermi liquid (or mean field "descendent" like the BCS state) strongly interacting fermions *have* to form quantum matter. The conclusion is that we have no clue how non-Fermi liquids work because their physics is shrouded behind the quantum supremacy [11] brick wall.

At this point the holographic duality of the string theorists kicks in. This is a quite serendiputous affair. The progress in string theory was driven by mathematics, aiming at the theory of quantum gravity. In the process the exceedingly elegant equations of the holographic duality were discovered. Confronting it with the empirical reality and especially very recently with help of quantum information it became increasingly clear that it just computes the phys-

ical properties of irreducibly many body entangled matter using the mathematical language of general relativity. The only worry is that it reveals a limiting case ("large N") which is in a poorly understood way "maximally" entangled. But limits have proven to be very useful in the history of physics and the hope is that holographic duality reveals ubiquitous principles governing the physics of compressible quantum matter. This in fact the marching order I wish to present as a theorist to the experimental community: use the condensed matter electrons as analogue quantum computer to check whether holographic duality delivers. We take this up at length in section 4.

## 3.2 Many body entanglement and unparticle physics

Twentieth century physics revolved around the particle idea. In the high energy realms particles are so ubiquitous that it landed in the name of the community studying it with accelerators. But it was also central to condensed matter physics: until the present day textbooks set out to explain the emergent elementary excitations that behave like quasiparticles which are much like the particles of the standard model. But in quantum physics the excitations are derivatives of the vacuum and it is actually very easy to understand that as necessary condition for finding particles in the spectrum, the vacuum has to be a SRE product, Eq. (4). Conversely, dealing with a many body entangled vacuum state there are no particles in the spectrum. In condensed matter language this is called "incoherent spectral functions" – "coherence" refers to the single particle quantum-mechanical coherent (wave-like) nature of the excitation. Hence, the presence of incoherent spectra is a diagnostic signalling that one may be dealing with quantum matter. As a caveat, such "unparticle spectra" may have bumps that seem to disperse as band structure electrons. The difference between "particle" and "unparticle" may be subtle since this is encoded in the details of the lineshapes. I will show an example in the below. Although there are ambiguities, the strange metals surely have an appetite to exhibit unparticle responses.

To understand the origin of particles as manifestation of the SRE product vacuum is very simple. However, it appears that it is not yet widely disseminated in the community at large and let me therefore present here a concise tutorial. First, what is an excitation of a quantum system? It all starts with the symmetry of the system, that defines the conserved quantities that are enumerated in terms of quantum numbers. In a typical condensed matter system these refer to total energy, crystal momentum, spin, charge and so forth. The vacuum state is characterized by such a set of quantum numbers. "Excitation" means that one inserts a different set of quantum numbers and pending the specifics of the measurement this translates into the probability of accomplishing this act, while these probabilities are collected in spectral functions. A particle is of course an excitation, but it is actually of a quite special kind. It can be described in a single particle basis where these quantum numbers are sharply localized, and this property requires a SRE product structure of the vacuum.

Let me illustrate this with a very simple example that is nevertheless fully representative: the transverse field Ising model. This model is a central wheel in Sachdev's book [32] while the remainder of this section is in essence a short summary of the central message of this book. Transversal field Ising describes a system of Ising spins with nearest neighbor interactions on a lattice, perturbed by an uniform external field with strength B in the $x$ direction,

$$H = -J \sum_{<ij>} \sigma_i^z \sigma_j^z - B \sum_i \sigma_i^x. \tag{5}$$

In one space dimension for nearest neightbor couplings one finds that the Euclidean action is identical to the 2D Ising model. This can be solved exactly. Let us follow Sachdev [32] in using it as a fruifly where matters can be computed exactly. There are pathologies associated with this integrability condition but these can be savely ignored in the present context.

The control parameter is $g = B/J$. When $g << 1$ the ground state is just the Ising ferromagnet polarized along the z-direction corresponding with a product ground state $\cdots | \uparrow\rangle_z | \uparrow\rangle_z | \uparrow\rangle_z | \uparrow\rangle_z \cdots$. Similarly, when $g >> 1$ the system is completely polarized in the x-direction with vacuum $\cdots | \uparrow\rangle_x | \uparrow\rangle_x | \uparrow\rangle_x | \uparrow\rangle_x \cdots$. Famously, when $g = g_c$ the system undergoes a genuine quantum phase transition ($QPT$) corresponding with the Ising universality class in $d + 1$ dimensions ($d$ is the number of space dimensions). When $g \neq g_c$ the vacuum is an SRE product departing from the classical vacua on either side of the QPT. Upon approaching the QPT the "classical" amplitude $A$ in Eq. (4) will steadily decrease to vanish precisely at $g_c$ where the many body entangled critical state takes over for $d = 1, 2$ (ignoring subtleties associated with integrability in $d = 1$).

The typical excitation spectrum is associated with the insertion of a spin triplet quantum number at an energy $E$ and crystal momentum $k$. For $g < g_c$ one encounters the specialty of one dimensional physics that this triplet fractionalizes in two kinks carrying both spin $1/2$ that propagate as independent particles – the spectrum associated with the triplet will appear incoherent although it is still controlled by the SRE product vacuum but this is a special effect of one dimensional physics although it also occurs in deconfining states of gauge theories in higher dimensions characterized by topological order. The generic situation is encountered for $g > g_c$ departing from the x-polarized ground state. Inserting a spin-flip relative to this ground state one finds invariably that the "bottom" of the spectrum realized a $k \to 0$ is of the form

$$G(k = 0, \omega) = \langle \sigma^z \sigma^z \rangle_{k,\omega} = \frac{A^2}{\varepsilon_{k=0} - \omega} + G_{incoh}. \tag{6}$$

The system will be characterized by a gap where $Im G_{incoh} = 0$ while inside this gap an infinitely long lived quasiparticle resides, signalled by the delta function at $\omega = \varepsilon_{k=0}$. For increasing $k$ the quasiparticle pole will disperse upwards acquiring a life time expressed by a conventional perturbative self energy. Upon approaching the critical point the spectral weight of the quasiparticle peak $\sim A^2$ will gradually diminish, becoming very small when $g \to g_c$. This is clearly a representative example of a typical quasiparticle.

Obviously, the spectrum is optimally particle-like in the limit $g \to \infty$ since the spectrum is completely governed by the delta function ($A^2 \to 1$). But this is just the classical limit where the vacuum is a pure product. The reason is obvious: injecting $\Delta S^x = \pm 1$ in the product vacuum flips the spin at one site. Together with its energy, the spin flip forms a lump that is *precisely* localized in position space: this is just the particle of classical physics. Upon switching on a small $J$ a back on the envelope calculation shows that this particle start to hop around, forming Bloch waves characterized by a dispersion $\varepsilon_k$. For increasing $J$ one has to work harder and harder to dress it up with perturbative quantum "corrections". In the quantum information language this means that the system develops many-body entanglement but this stops at an entanglement length which is in this case coincident with the correlation length associated with the quantum phase transition. But at length scales larger than this entanglement length the system behaves as if $J \to 0$: it becomes again the quantum mechanical particle but now with an internal structure governed by SRE. The overlap with the "bare" spin flip is then set by $A$, explaining the behavior of the pole strength. One recognizes the general structure of semi-classical field theory that is intuitively assumed in the textbooks.

What happens precisely at the quantum critical point? One just needs to know that an emergent scale- and Lorentz invariance is realized precisely at this point; in fact, conformal symmetry takes over and this is much liked by string theorists because it is a very powerful symmetry that greatly simplifies the math. An example is the principle that two-point propagators are "completely governed by kinematics". This rests on the simple wisdom that in scale invariant systems all functions have to be power laws and it is very easy to see that this

implies [7,32]

$$G(k, \omega) = = \frac{1}{\sqrt{c^2 k^2 - \omega^2}^{2\Delta_\sigma}}, \tag{7}$$

where $c$ is the emergent "velocity of light" and $\Delta_\sigma$ is the "anomalous" scaling dimension of the operator $\sigma_x$. This is some real number that is associated with the universality class that has to be computed: it is the same thing as the correlation function exponent of the Wilson-Fisher renormalization group of statistical physics. How does the spectral function looks like? Take $k = 0$ and the imaginary part $\sim 1/\omega^{2\Delta}$: this is just a powerlaw in frequency. Upon boosting this to finite momentum the spectral function vanishes for $\omega < ck$ while it turns in the powerlaw at higher frequency. This results in the "causal wedge"; to convince yourself that holography gets this automatically right have a look at Fig. 2A in ref. [34].

The issue is that although the spectral function is quite dispersive, showing peaks at $\omega = ck$, the analytical form of the branch-cut Eq. (7) is actually very different from the "particle poles" Eq. (6). The branch cut reveals "unparticle physics": it is just impossible to identify a particle, let alone to use it as construction material for computations. It cannot be emphasized enough how misleading the very idea of a particle is under these circumstances. This becomes crystal clear referring back the many body entangled ground state Eq. (3). The eigenstates associated with the insertion of the quantum numbers are of the same type: it is impossible to locate them in single particle position space. Much worse than that, these are delocalized in the enormous many body configuration space! Literally, "everything knows about everything" and when one throws in something everything is altered. Entanglement becomes truly spooky in field theory! No wonder that there are no particles in the spectrum and we are just saved by conformal symmetry dealing with conventional quantum critical states, constraining greatly the outcomes.

### 3.3 Unitary time evolution versus thermalization

To conclude this tutorial on elementary notions of quantum information, there is yet another enlightment originating in quantum information that should be thoroughly realized since it is crucial for the appreciation of what follows. There is quite some mathematical machinery here that I will not address. The central nave is the "eigenstate thermalization hypothesis". This ETH idea was formulated in the early 1990's in the quantum information community [35] but it took quite a long while before it was appreciated in the physics community at large [12]. Ironically, the string theorists were uncharacteristically in the rearguard. Some two years ago it started to sing around in this community, turning rapidly into the next hot idea, actually for a good reason. Some firm evidence was found establishing holography as a machine computing quantum thermalization of extreme many body entangled matter. The core subjects of this primer – rapid hydrodynamization, Planckian dissipation and minimal viscosity – are at least in holography manifestations of such physics and the call to experiment is just to find out whether strange metal electrons are sufficiently many body entangled to give in to these new principles.

Let us start with a most elementary quantum information question: why does a quantum computer process information? One prepares an initial state $|\Phi(t = 0)\rangle$ to then switch on the quantum circuitry that has the action to unitarily evolve this state in time: $|\Phi(t)\rangle = e^{iHt}|\Phi(t = 0)\rangle$. Profiting from the large "$2^N$" many bit/body Hilbert space, one discerns that this "exponential speed up" is a "computational resource" having the potential to solve NP hard problems. But there is a serious problem. Consider the total von Neumann entropy of the system: $S(t) = Tr[\rho_{tot}(t) \ln \rho_{tot}(t)]$, where $\rho_{tot}$ is the *full* density matrix of the system containing all states, $\rho_{tot}(t) = \sum_{n=1,2^N} |\Psi_n(t)\rangle\langle\Psi_n(t)|$. It is very easy to show that this entropy is stationary under the unitary time evolution, $dS/dt = 0$. So what? The trouble is that

this entropy has the same status as a Shannon entropy in a classical system, keeping track of information [27]. That it is stationary implies that *no information is processed*: as long as the quantum computer is doing its quantum work it is not computing! In order to compute the wave function has to collapse, called the "read out", and this yields the probability for a particular string of classical bits to be realized. The tricky part is then to show that this readout information suffices to profit from the exponential speed up [36]. Shor's algorithm for prime factorization is the canonical example.

But nature works in the same way! The microscopic constituents are supposed to be subjected to a unitary time evolution. For whatever reason, wave functions collapse and when they do so something happens: nature "computes". ETH is just claiming that in the absence of the extremely delicate control that the quantum engineers hope to accomplish there is just one outcome. The "read out" will come to us who can only build machines that probe reality "on the dark side of the collapse" as if the state has just turned into an *equilibrium thermal state* when we wait long enough. I am prejudiced that the allure of ETH in the physics community may have its origin in a quite human subconcious condition. I am sure that most of us share the experience that you tried once to impress a non-physicist as part of a courtship process with your latest insights in black hole physics, whatever. You may remember the outcome: in no time that person was thinking that you were very drunk, severely stoned or likely an autistic nerd. ETH is just the asymptotic form of this phenomenon: $2^N$ numbers are changing in a highly orchestrated fashion but you can keep only track of order $N$ outcomes and as a victim of this overload of information you interpret it as chaos, randomness, entropy production.

ETH makes it precise. The classic formulation is as follows [12,35]: depart from an initial state that is pure, in the form of a coherent superposition of excited densely entangled energy eigenstates narrowly distributed around a mean energy: $|\Psi(t = 0)\rangle = \sum_n a_n |E_n\rangle$. A caveat is that in a finite energy interval there will be an exponential number of energy eigenstates in the superposition given that these are all densely entangled. Let this state unitarily evolve in time such that the state remains pure at later times. Any physical observable (after the collapse) has the status of being an expectation value of a "local" operator $\hat{O}$: "local" refers to an operator that only involves a logarhitmically small part of the many body Hilbert space, and this represents any measurement that mankind has figured out. The hypothesis is formulated in a precise way as follows,

$$\langle \Psi(t)|\hat{O}|\Psi(t)\rangle = Tr\left[\rho_T \hat{O}\right] \tag{8}$$

at a sufficiently long time $t$, while $\rho_T = \sum_n e^{-E_n/(k_B T)}|\Psi_n\rangle\langle\Psi_n|$, the thermal density matrix associated with the temperature $T$ of the equilibrium system set by the total energy that was injected at the onset. This is just the surface; the case can be made surprisingly precise by involving just some notions of the "typicality" of the states. For instance. ETH fails for integrable systems because of the infinity of conservation laws and instead one finds a non-thermal "generalized Gibbs ensemble" [12].

It is a scary notion. Even when you light a match you are supposed to fall prey to the overload of quantum information. However, dealing with weakly interacting matter like the gas you are breathing it turns out that ETH just reduces to the textbook story that at high temperatures the atoms just turn into tiny classical balls that are colliding against each other. The "classical delusion" is just a perfect representation of the physics: for lively illustrations, see e.g. ref. [13,37] However, dealing with the densely entangled, strongly interacting matter of unparticle physics it may happen that the quantum thermalization produces outcomes that *cannot be possible mapped onto an analogue classical system*.

The case in point is the Planckian dissipation. I already highlighted the "unreasonable simplicity" of relaxation times that only know about $\hbar/(k_B T)$ from a semi-classical (particle physics) perspective. It is a rather recent development to conceptualize it in an ETH language

but this is to quite a degree a matter of technical language. ETH is typically formulated in the language of canonical quantum physics, revolving around Hamiltonians and wave functions. On the other hand, Planckian dissipation was identified employing the equivalent Euclidean field theory formalism. This is the affair that the path-integral description turns into an effective statistical physics problem dealing with "bosonic" problems, in a $d + 1$ space where the extra dimension is imaginary time. As first realized by Chakravarty *et al.* [38] and further explored by Sachdev [32], $\tau_\hbar$ follows in fact from elementary scaling notions applied to the critical state of statistical physics. In this formalism the dynamical *linear response* functions are obtained by analytic continuation from the "stat. phys." Euclidean correlation functions to real time. In canonical language these have the "VEV of local operator" status "around" equilibrium.

In the Euclidean formalism it is very easy to address finite temperature: the imaginary time dimension is compactified in a circle with radius $R_\tau = \hbar/(k_B T)$. At the quantum critical point a stat. phys. critical state is realized in Euclidean space time and when this is strongly interacting (many body entangled) *hyperscaling* is in effect: the response to finite size becomes universal. Since the Euclidean space-time system has become scale invariant, the only scale is the finite "length of the time axis" when temperature is finite and it follows that $R_\tau$ is the only scale in the system. The only surprise is that after analytic continuation to real time this time scale acquires the status of a *dissipative* time, $\tau_\hbar$. This was a bit mysterious until we realized that this is just ETH at work in the densely entangled scale invariant quantum state: in the Euclidean formulation it just becomes very easy to understand why the outcome is so simple.

The string theorists knew all along about Planckian dissipation but they found it so obvious that they did not bother to give it a name. Surely, I was myself overly aware of the Euclidean story when I drew the attention to this Planckian time being at work in the resistivity [15] as explained in section 2. However, there is still quite a long way to go in order to explain why this time governs transport in the way it does. There are two complications: (a) The arguments of the previous paragraph only apply to *non-conserved* order parameters such as the staggered magnetization. Dealing with electrical transport one runs into conversation laws having in general the effect that $\tau_\hbar$ will enter indirectly in the current response. (b) As will be explained next, the strange metal state appears to be not scale invariant (conformal). Instead, it should be *covariant* under scale transformation for reasons that were identified by the holographists. More powerful machinery is needed: the black holes of the string theorists.

# 4 The strange metals of holography

The new mathematical kid on the block in this context is the AdS/CFT correspondence or "holographic duality". The "correspondence" is by far the most important mathematical machine that came out of a math-driven pursuit that has been raging for about 40 years. Since its discovery in 1997 [39] it has become the central research subject in what used to be the string theory community – string theorists as of relevance to condensed matter prefer to be called "holographists". The correspondence evolved in a bigger than life mathematical "oracle" that has the strange capacity to unify all of known physics producing in the process quite some unknown physics. Several excellent books have been written [6, 8, 40] and for the purpose to find out how it relates to condensed matter one has to study the $\sim 600$ pages of our recent textbook [7]. In order to follow the arguments that motivate the linear resistivity predictions let me present here a minimal "tourist guide", not explaining anything really but hopefully serving the purpose that the reader has some idea where it comes from. First I will present a very short summary of AdS/CFT generalities, to then zoom in on the condensed matter application that matters most: the holographic strange metals. I present here a view inspired by ideas of

Gouteraux that is not quite standard. At the least metaphorically, I find it convenient to view the holographic strange metals as densely entangled generalizations of the Fermi liquid. I am looking forward to debate this further with professional holographists. I then introduce the standard lore of elementary holographic transport theory focussing on the minimal viscosity. In the final subsection I will highlight the confusing affair related to the fate of the minimal viscosity in the strange metals.

## 4.1 How it started: AdS/CFT, at zero density

Let's first get an impression of the meaning of the acronym AdS/CFT, the plain vanilla version where it all started [39]. CFT stands for "conformal field theory" and this is just another name for the familiar bosonic quantum critical affair. This has a long history among string theorists, given that conformal invariance adds quite some power to the math. But compared to the simple spin models of condensed matter physics there is much more going on. It turns out that maximally supersymmetric field theories have the attitude that one does not need to tune to critical points: such theories are automatically critical over a range of coupling constants. The microscopic ("UV") degrees of freedom are those of Yang-Mills theory and to get the correspondence working one has to take the limit that the number of colors ($N$) goes to infinite, together with the " 't Hooft" coupling constant. This is so called matrix field theory, and different from the vector large $N$ limit (familiar from slave theories) this limit describes a "maximally" strongly interacting quantum critical state. Different from the 1+1D CFT's, these large central charge theories living in higher dimensions are not at all integrable. There are various way to argue that these are in one or the other way maximally entangled.

AdS is the acronym for "Anti-de-Sitter" space. This is just the geometry that follows from general relativity ("Einstein theory", "gravity") for a negative cosmological constant. It has the role of ubiquitous brain teaser in graduate GR courses aimed at highlighting the weirdness of curved space time. It is an infinitely large place that still has a boundary, while it takes only a finite time to get from the boundary all the way to the middle ("deep interior"). The game changer was the discovery of Maldacena in 1997 that gravity in AdS in one higher dimension is "dual" to the CFT. The word "dual" has a similar sense as in "particle-wave duality". There is really one "wholeness" (quantum mechanics) but pending the questions one asks one gets to see particles or waves, "opposites" that are related by a precise mathematical relation (the Fourier transform). The correspondence relates in a similar way classical gravity to the extremely quantal CFT, where the role of the Fourier transform is taken by an equally precise "Gubser-Klebanov-Polyakov-Witten" (GKPW) rule on which the so-called "dictionary" rests that specifies in a precise way how to translate the quantum physics into gravity and vice versa. It is called "holographic" since the gravitational side has one extra dimension: this "radial direction" connects the boundary to the deep interior and has the identification as the scaling direction in the field theory. The claim is that AdS/CFT geometrizes the renormalization group and upon descending deeper in AdS one "sees" the physics at longer times and distances. The deep interior codes for the macroscopic scale ("IR").

The correspondence is generally regarded as being mathematically proven at least to physics standards, with the caveat that it is only useful in the large N limit because for finite N the bulk is governed by poorly understood stringy quantum gravity. The latest developments rest on the cross-fertilization with quantum information: the "its from quantum bits" program. This refers to an intriguing slew of developments having as common denominator that the geometry of the bulk is actually associated with the way that the entanglement is stitched together in the vacuum state of the boundary. Surely, the correspondence mines ETH information – VEV's of local operators such as two point functions – and now the unreasonable simpicity is encoded in the geometry. AdS is among the most simple GR geometries but it does accurately encode for the simplicity of the unparticle responses; e.g., the two point functions are neat

branch-cuts. It also applies to the finite temperatures where the motives behind the Planckian dissipation are hard wired.

How to encode the finite temperature in the boundary in terms of bulk geometry? The universal answer is: in the form of a black hole inserted in the deep interior. Remarkably, the entropy of the boundary is coincident with the Bekenstein-Hawking black hole entropy which is set by the area of the horizon while the temperature is equal to the Hawking temperature. In AdS gravity tends to work differently than for flat asymptotics; for instance, the temperature increases when the black hole gets bigger. We will see soon that there are direct relations between Planckian dissipation phenomena and black hole physics.

## 4.2  Finite density: strange metals and other holographic surprises

This "plain vanilla" AdS/CFT is not so greatly useful for condensed matter purposes since it does not deliver that much more as to what is already explained in Sachdev's textbook [32]. But this changes radically gearing the correspondence to address *finite* density physics. This is encoded by the presence of an electrical monopole charge in the deep interior, having the effect that the gravity becomes much richer. By just pushing the GR in the bulk, a physical landscape emerges in the boundary that is suggestively similar to what is observed in cuprates [7]. In fact, the latest results indicate that this GR has a natural appetite to reproduce the intricacies of the interwined order observed in the pseudogap regime for the reason that the relevant black holes have to be dressed with quite fancy black hole "hair" [41, 42]. But here we are interested in the strange metal regime.

Finite density in the boundary is dual to a charged black hole in the bulk [7]. The near horizon geometry of such black holes, representing the IR physics in the boundary, is an interesting GR subject giving rise to surprises. A minimal example is the Reissner-Nordstrom (RN) black hole known since the 1920's which was the first one looked at in holography [43]. Its near horizon geometry turns out to be $AdS_2 \times R^d$. This describes a metal in the boundary characterized by *local quantum criticality*: only in temporal regards the system behaves as a strongly interacting quantum critical affair. This caught directly my attention since it was well known [20] that this local quantum criticality seemed to be at work in the cuprate strange metals (see ref. [44] for very recent direct evidence). One can express it in terms of the dynamical critical exponent expressing how time scales relative to space: $\tau \sim \xi^z$. For an effective Lorentz invariant fixed point $z = 1$; in a Hertz-Millis setting $z = 2$ dealing with a non-conserved order parameter (diffusional due to Landau damping), while the maximal $z = 3$ is associated with a conserved ferromagnet. Local quantum criticality means $z \to \infty$ which is very hard, if not impossible to understand in the Wilsonian paradigm underlying the bosonic quantum criticality. Holography surely knows about fermions, and apparently reveals here genuinely new "signful" quantum matter behavior.

RN is just the tip of the iceberg. This is the unique black hole solution for a system that only knows about gravity and electromagnetism. However, the string theoretical embedding of the correspondence insists that one also should incorporate *dilaton* fields. These are an automatic consequence of the Kaluza-Klein reduction inherent to the construction. It turns out that the near horizon geometry of such "Einstein-Maxwell-Dilaton" (EMD) black holes can be systematically classified [45]. These describe a scaling behavior of the boundary strange metals involving next to $z$ also the so-called "hyperscaling violation exponent" $\theta$ . This sense of hyperscaling is detached from the meaning it has in Wilsonian critical theory where it refers to what happens to the order parameter approaching the critical point. These strange metals are behaving as *quantum critical phases* [7] in the sense that they are not tied to a quantum critical point. The near horizon geometry determines the thermodynamics in the boundary and hyperscaling now refers to the way that the thermodynamically relevant degrees of freedom scale with the volume of the system. Another side of this affair is that these strange metals

are not *invariant* under scale transformation (as the zero density CFT's) but instead they are *covariant* under scale transformations as can be easily seen from the form of the near-horizon metric [1].

I percieve this quantum critical phase behavior as predicted by holography as an inspiration for perhaps the most enlightened question one can pose to experiment. Upon observing algebraic responses the community jumped unanomously to the conclusion that this should be caused by an isolated quantum critical point associated with some form of spontaneous symmetry breaking coming to an end. There was just nothing else that could be imagined. However, for many years the debate has been raging regarding the identification of this very powerful order parameter, without any real success. Arguably, there may be even empirical counter-evidence, ruling out the quantum critical point [46].

The holographic strange metal is completely detached from such a Wilson-Fisher critical state. There is actually one familiar state of matter that exhibits this kind of scaling behavior: the Fermi liquid! Is a Fermi-liquid a critical state- or either a stable state of matter? This question has been subject of quite some debate. On the one hand, it can be argued that the Fermi liquid behaves like a cohesive state that is actually remarkably stable – the critical state associated with a quantum critical point is singularly unstable, any finite perturbation will drive it to a stable state. However, all physical properties are governed by power laws: the resistivity is quadratic in temperature, the entropy grows linear in temperature and so forth. Both sides of the argument are correct: the algebraic responses signal that scale transformations are at work while at the same time the Fermi energy protects the state. It is actually a semantic problem. Quantum *critical* means that the deep IR state has no knowledge of any scale (other than temperature) whatsoever, and the emergent state is truly *invariant* under scale transformation. But not so the Fermi liquid: the collective properties do remember the Fermi energy. The algebraic responses always involve the ratio of the measurement scale (temperature, energy) and the Fermi energy. The collision time of the quasiparticles can be written as $\tau_c \simeq (E_F/k_B T)\tau_\hbar$: the "scale invariant" Planckian time is modified by a factor $(E_F/k_B T)^{\# = 1}$. The entropy in a *conformal* (quantum critical) system is set by $S \sim T^{d/z}$ just reflecting the finite size scaling of the conformal system. For $z = 1$ (Lorentz invariance) one recognizes the Debye law $S \sim T^d$. The Fermi liquid is governed instead by the Sommerfeld entropy $S \sim (k_B T/E_F)^{\# = 1}$. This is the meaning of being "covariant under scale information." The stability of such a state is now governed by the "scaling dimensions of operators." Consider the dynamical susceptibility associated with a physical quantity: this will behave as $\chi(\omega) \sim (\omega/E_F)^\#$. When the scaling dimension $\# > 0$ it is "irrelevant" and it will just die out while for $\# < 0$ the response will diverge signaling that the Fermi-liquid will get destroyed. The stability of the Fermi liquid is associated with the most "dangerous" operator being the "marginal" pair operator with exponent $\# = 0$; this translates in a real part $\chi'(T) \sim \log(T/E_F)$, the "BCS logarithm".

The best way to conceptualize the strange metals of holography seems to be to view them as "strongly interacting" generalizations of the Fermi liquid, in the critical sense of the word [2]. Consider the conventional, bosonic critical state; above the upper critical dimension this is governed by the Gaussian fixed point meaning that the critical modes are described by free fields. In the quantum interpretation these become particles again and even the quantum critical state is no longer many body entangled. The scaling dimensions become all simple (ratio's of) integers. Below the upper critical dimension the NP-hardness kicks in, the quantum critical state is many body entangled, and the scaling dimensions become anomalous: arbitrary real numbers that in turn render the response functions to become the "unparticle" branch cuts. The Fermi liquid is like the Gaussian critical state, characterized by product state structure, particles in the spectrum and integer scaling dimensions. Holography suggests that

---

[1] At the time of writing the book we were not aware of this: I am grateful to Blaise Gouteraux for explaining it.

[2] This is not resounding a community consensus, but I do find it myself a quite comfortable view on this mystery.

this can be generalized in the form of a state having the same covariant scaling behaviour but now characterized by anomalous dimensions which in turn signal that the vacuum is densely entangled.

To see how this works, let us zoom in on the thermodynamics. I already announced the $z$ and $\theta$ exponents which are the only scaling dimensions that can be introduced for the thermodynamics, departing from the covariant scaling principle. It turns out that the entropy scales according to

$$S \sim T^{(d-\theta)/z}. \tag{9}$$

Surely, this includes the zero density conventional quantum critical point case where $\theta = 0$. To get an intuition for $\theta$, let us consider the Fermi-liquid. It seems quite natural that entropy should know about the dimensions of space, but this is not at all the case for the Fermi-liquid. According to the Sommerfeld law $S \sim T/E_F$ regardless the dimensionality of space $d$. The way to count it is as follows. Start in $d = 1$: the Fermi-surface is pointlike, and a linear branch of excitations departs from this point, implying the Lorentz invariant $z = 1$; this scales in the same way as a "CFT$_2$", a conformal state in 1+1D with $S \sim T$. But in 2 space dimensions the Fermi surface becomes a line while at every point on the line the excitations scale like a CFT$_2$, and therefore again $S \sim T$. This repeats in higher dimensions: the bottom line is that $\theta = d - 1$ (dimension of the Fermi surface) while $z = 1$ such that $S \sim T$ always.

But we argued in the above that there is somehow "local quantum-critical like" stuff around in the deep IR characterized by $z \to \infty$. Also in a "conventional" marginal Fermi liquid (MFL) setting [20] one has to face the impact of this stuff on the thermodynamics: in this context it is in first instance introduced as a heat bath dissipating the quasiparticles, but these same states [44] of course contribute to the entropy. This is worked under the rug in the standard MFL story, where one only counts the dressed quasiparticles.

Assuming that there is still a Fermi surface ($\theta = d - 1$) it follows that $S$ should become temperature independent when $z$ is infinite. This is manifestly inconsistent with the data: at the "high" temperatures where the strange metal is realized electronic specific heat measurements show Sommerfeld behavior, $S \sim T$. But Eq. (9) shows yet another origin of a Sommerfeld entropy. One should avoid a temperature independent contribution since this implies ground state entropy (this is actually going on in the primitive RN metal). However, by sending $\theta \to -\infty$ keeping the ratio $\theta/z$ fixed at the value $-1$ one finds yet again the Sommerfeld law!

Although it is far from clear how to interpret such an infinitely negative $\theta$, employing the big string theoretical machinery behind holography [6] one can show that a "physical" theory exists exhibiting this behavior. Up to this point I have been discussing results of the "bottom-up" approach. One just employs the principles of effective field theory in the bulk, not worrying about the precise form of the potentials to arrive at the generic form of the scaling relations where the scaling dimensions are free parameters. There are bounds on these scaling dimensions but these are weak: $z \geqslant 1$ because of causality, while $\theta < d$ for thermodynamic stability. However, employing full string theory one can derive specific holographic set ups where one can identify the explicit form of the boundary field theory characterized by specific scaling dimensions. As it turns out, the $\theta/z = -1$ example I just discussed which is called "conformal to AdS$_2$" [47] has such a "top-down" identification [48] and it therefore represents a physical choice of scaling exponents. Surely, this top down theory is yet again of the large N kind having nothing to do with the "chemistry" of the electrons in cuprates. However, one may wish to just take the scaling principles at face value, to continue phenomenologically. We learn from experiment that $z \to \infty$, and in combination with the Sommerfeld specific heat the scaling principle Eq. (9) leaves no room to conclude that one is dealing with a many body entangled state characterized by $\theta \to -\infty$.

In the holographic calculation that gave us the idea for the linear resistivity explanation we actually employed the conformal to $AdS_2$ metal. Perhaps the real take home message of this section is that it illustrates effectively the track record of holography in condensed matter physics. It just seem to excell in teaching mankind to think differently. Listening to the mumblings of this oracle with its strange supersymmetric large N feathers one discovers that it is quite easy to expand one's views on what is reasonable phenomenological principle. In fact, quite recently a lot of work was done on the so-called Sachdev-Ye-Kitaev model [49] which is an explicit strongly interacting "signful" quantum model that is nevertheless tractable. It is effectively a 0+1 D quantum mechanical problem, involving a large number ($N$) of Majorana fermions interacting via random interactions. In a highly elegant way it has been demonstrated that its emergent deep IR is precisely dual to a Reissner-Nordstrom black hole in $AdS_2$ [52]: proof of principle that strange metal physics can emerge from condensed matter like UV circumstances.

Although a sideline in the present context, the holographic metals do share the property with Fermi liquids to become unstable toward symmetry breaking at low temperatures. Given that such a metal itself may be viewed as a generalization of the Fermi liquid, the mechanisms of holographic symmetry breaking may be viewed as a generalization of BCS. It is about the algebraic growth (relevancy) or decline (irrelevancy) of the operator associated with the order as function of scale. Before we had realized how this works in holography we had already anticipated the result on basis of a mere phenomenological scaling ansatz: the "quantum critical BCS" [50]. Instead of the marginal scaling dimension of the pair operator of the Fermi gas we just asserted that this may just be an arbitrary dimension. When it is relevant the outcome is that even for a weak attractive interactive interaction $T_c$'s may become quite large. As it turns out, this is quite like the way that holographic superconductivity works [51]. The latest developments reveal an appetite for this holographic symmetry breaking to produce complex ordering patterns that share intriguing similarities with the intertwined order observed in the underdoped cuprates [41,42].

## 4.3 Dissipation, thermalization and transport in holography

Transport phenomena have been all along very much on the foreground in holography. This is in part because of historical reasons as I will explain next. Another reason is that the mathematical translation of the bulk GR to DC transport in the boundary turns out to be particularly elegant [7,8]. It is also of immediate interest to the condensed matter applications since it is perhaps the most obvious context where holography suggests very general principle to be at work. In a very recent development evidences are accumulating that these are just reflecting general ramifications of ETH (section 3.3) in exceedingly densely many body entangled matter.

The application of AdS/CFT to the observable universe jump started in 2002 when Policastro, Son and Starinets [4] realized that the finite temperature, macroscopic CFT physics is governed by relativistic hydrodynamics, with a dissipative side encapsulated by a shear viscosity having a *universal* dual in the bulk. This is very elegant. We learned that the finite temperature state in the boundary is dual to a Schwarzschild black hole in the deep interior of the bulk. As it turns out, the viscosity $\eta$ is just equal to the *zero frequency absorption cross section of gravitational waves by the black hole*. According to a GR theorem this cross section is just set by the area of the black hole event horizon. But the entropy density in the boundary $s$ is coincident with the Bekenstein-Hawking entropy which is also set by the horizon area. Taking the ratio of the two there is just a geometrical factor of $1/(4\pi)$ and the big deal is that

$\hbar$ sets the dimension,

$$\frac{\eta}{s} = \frac{1}{4\pi}\frac{\hbar}{k_B} \tag{10}$$

as it turns out, compared to normal fluids this ratio is extremely small [5]; a debate has been raging whether it may represent an absolute lower bound. It was therefore called the "minimal viscosity", with the caveat that $\eta$ itself may be quite large in a system with large entropy density. It made headlines in 2005. After much hard work at the Brookhaven relativistic heavy ion collider (RHIC) definitive evidence appeared in 2005 that they had managed to create the *quark-gluon plasma* [3,40] and to their big surprise this turns out to behave as a hydrodynamical fluid with a viscosity which is remarkably close to Eq. (10)! This was more recently claimed to be confirmed in a cold atom fermion system, tuned to the unitarity limit where it also turns into a quantum critical system [53].

This surprise was spurred by the theoretical anticipation based on perturbative "particle physics" QCD predicting a ratio that is many orders of magnitude larger than the minimal one [5]. In hindsight, it looks like that the viscosity is just one of these quantities were the difference between semiclassical thermalization and the quantum thermalization in densely entangled matter becomes very pronounced. This can be discerned using elementary dimensional analysis. Let us first consider a thoroughly studied condensed matter example: $^3$He in the finite temperature Fermi liquid regime. As everything else living in the Galilean continuum at finite temperature, this behaves macroscopically as a Navier-Stokes fluid. According to kinetic theory describing the weakly interacting quasiparticles, the viscosity is set by the free energy density $f$ and the momentum relaxation time $\tau_K$, in the spirit of dimensional analysis [7]: $\eta \simeq f \cdot \tau_q$ where $\simeq$ is associated with dimensionless numbers of order unity. The momentum relaxation time is the usual $\tau_q = (E_F\hbar)/(k_BT)^2$ while the free energy density $f = E_F$ when $k_BT/E_F << 1$. It follows that $\eta = \hbar(E_F)^2/(k_BT)^2$, this turns out to be an accurate estimate of the order of magnitude of the viscosity, while the $1/T^2$ dependence on temperature is correct [54]. The conclusion is that at low temperatures the Fermi-liquid fluid becomes *extremely* viscous, a well documented technical complication for the construction of dilution fridges. What is the reason? The quasiparticles fly freely over increasingly large distances when temperature is lowered before colliding against other quasiparticles. Thereby the momentum they carry is transported over longer and longer distances and this translates into a very large viscosity.

Let us now turn to the finite temperature CFT living in the Galilean continuum and try out the same dimensional analysis. In order to encounter hydro the system should be in local equilibrium: the momentum relaxation time should be therefore coincident with the linear response relaxation time. Dealing with the many body entangled critical state we expect from the rapid quantum thermalization that any relaxation time should be Planckian, $\tau_q \simeq \tau_\hbar$. In a scale invariant quantum state there cannot be internal energy since it is a scale, and therefore the free energy density is entirely entropic: $f = sT$. The viscosity follows as: $\eta \simeq f \cdot \tau_q \simeq sT \cdot \hbar/(k_BT) = s(\hbar/k_B)$. The minimal viscosity Eq. (10) is recognized: it is just a consequence of Planckian dissipation [7].

Another lesson from the quark gluon plasma physics that has relevance for the strange metal transport is the phenomenon called "rapid hydrodynamization". The heavy ion collision process is an extreme non-equilibrium affair. Two lead nuclei are slammed into each other with a speed near the velocity of light. In the ensuing fire ball the quark gluon plasma is formed. I just argued that there is convincing experimental evidence that the QGP behaves as a hydrodynamical fluid and here is the mystery. Hydrodynamics requires local equilibrium and this sets in typically after hundreds of collisions between the particles in a semi-classical system. The experimental claim is that in the time it takes for a quark to traverse a distance

roughly equal to the proton radius local equilibrium is already established [3]! Although completely incomprehensible from a semi-classical perspective, holographic toy models mimicking collisions do exhibit a similar fast hydrodynamization [3, 40].

In fact, a clue is offered by holographic set ups that model a typical condensed matter experiment: optical pump-probe spectroscopy [14]. The system is excited by an intense but very short laser pulse, and the time evolution of this excited state is then followed by linear response means such as the optical conductivity. This is actually a quite literal realization of the ETH protocol discussed in section 3.3. As it turns out, in holography for quite universal GR reasons *any causal measurement will reveal instantaneous thermalization*: it does not take any time after the pulse is switched off to reach equilibrium. This is the case both at zero- and finite density and the instantaneous thermalization is even explicitly identified in the large N CFT itself [55], as well as in the SYK model [56]. The conclusion is obvious: the ETH principle is just insisting that if one waits long enough the quantum evolution will end in a thermal state according to the local observer. However, dealing with non-equilibrium physics of the "maximally" entangled matter described by large N CFT's and holographic strange metals one expects the quantum thermalization to reach its extreme with the outcome that it does not take any time at all to thermalize! It is surely a limiting case, and the issue is whether matter that is suspected to be many body entangled will exhibit just very fast (instead of infinitely fast) thermalization behavior.

These are the lessons of the quark-gluon plasma which are of special significance to the *particular* holographic hypothesis for the linear resistivity that I would like to see tested experimentally. There is however much more: holography inspired transport theory developed in a vast subject by itself, in fact dominating the whole condensed matter portfolio. This whole affair was triggered in 2007 by Sachdev teaming up with holographists to address a magneto-transport phenomena (Nernst effect) in quantum critical fluids [57] – this is closely related to the material in the next sections. But it diversified since then in quite a number of directions, fuelled by elegant mathematical constructions that simplify the bulk computations considerably. Among others, this addresses the strong disorder limit [21] and very general scaling considerations that even get beyond holography [58]. This came to fruition after the deadline of our book and I recommend the more recent ref. [8] as the best source for this vast literature that is presently available.

The special merit of the next sections lies in the fact that it is in absolute terms the simplest idea to be found in this rich portfolio of holography inspired transport theories. But there is yet a deeply confusing side to it one should be aware off before we address the business end.

## 4.4 The caveats: some reasons for the minimal viscosity to fail in cuprates

The remaining sections of this paper revolve around the hypothesis that the principle of minimal viscosity is generic to a degree that it is even governing the transport in the cuprate strange metals. Let me emphasize again that it is no more than a hypothesis that ought to be tested by experiment. The available theory falls short in delivering an unambiguous answer – a quantum computer is required.

The minimal viscosity can be regarded as thoroughly tested under the conditions realized in the heavy ion collisions. In fact, the microscopic physics of the QCD quark-gluon plasma is in a number of regards quite different from the large N, supersymmetric UV of the correspondence suggesting that the minimal viscosity is a strong emergence phenomenon that is not overly sensitive to the nature of the microscopic degrees of freedom. Implicit to the discussion in the above, the guts feeling is that all what is required is dense entanglement and scale invariance.

It is yet a big leap to the electrons in cuprates. Departing from the strongly interacting particle physics inside the unit cell the fermion sign brick wall is standing in the way to follow the scaling flow towards the deep infrared. It is just guesswork what is going on at the other

side of the brick wall: the idea is that the planckian dissipation/minimal viscosity may be generic to densely entangled matter to a degree than it may even rule the cuprate electron system.

Within the specific context of cuprate strange metal further issues arise that will be discussed in the final section. However, even within the confines of AdS/CFT there are reasons to be less confident whether the minimal viscocity has such powers. As compared to e.g. the quark gluon plasma there are two gross differences with the strange metals: (a) the galilean invariance is badly broken in the UV by the presence of a large lattice potential, and (b) the strange metals are realized at finite density. Let me summarize what AdS/CFT has to say about their effects on the minimal viscosity.

### 4.4.1 Umklapp scattering versus the viscosity

Let us first zoom in on the effects of the background lattice. Although not emphasized in solid state physics courses, the Fermi liquid living in a strictly periodic potential has the remarkable property that Galilean invariance is restored in the deep IR. It is simple: in the absence of quenched disorder the zero temperature residual resitivity vanishes in a normal metal. The absence of resistivity implies that total momentum is conserved and this in turn requires that Galilean invariance is restored. This is of course understood: the quantum mechanical quasi-particles live in momentum space and are thereby delocalized in real space, averaging away the inhomogeneity associated with the lattice. In momentum space this boils down to the decoupling of the quasiparticles from the large Umklapp momenta. When the perfect periodicity is interrupted by quench disorder, scattering associated with small momenta switches on and this causes the residual resistivity to become finite.

It is conceptually straightforward to include background lattices in AdS-CFT [7,8]. Interestingly, the outcome is that rather generically holographic strange metals share the property with the Fermi liquid that "Umklapp is irrelevant": the periodic potential disappears in the deep infrared. At low temperatures a hydrodynamical fluid can therefore be formed. In turn, one can subsequently add weak disorder to study the "near hydrodynamical regime". This will be the point of departure in the next sections.

The lattice will break the translational invariance as well in the gravitational bulk having the effect of greatly complicating the bulk gravity. One has now to solve systems of non-linear partial differential equations that can only be achieved numerically. Although gross features like the irrelevancy of the lattice potentials can be easily deduced from the numerical solutions, the precise nature of the near horizon geometry encoding for e.g. the viscosity is a more subtle affair that is still rather poorly understood. The holographists constructed various patches to study aspects of this problem which are all departing from a homogenous space, wiring in various forms of total momentum sinks [8]. These are the "linear axions", "Q-lattices" and "massive gravity". These have nothing to say about Umklapp scattering per se, requiring inhomogeneous spaces by default. With the caveat that these are given in by mathematical convenience, it is believed that these are nevertheless representative for the long wavelength physics associated with real lattices. Dealing with weak disorder these all yield the same results and the holographic set up of the next section rests on the minimal massive gravity trick.

The question arises, what happens with the $\eta/s$ ratio in a holographic system characterized by a large periodic potential in the UV that is irrelevant in the IR? In the infrared momentum is conserved again and viscosity becomes meaningful. However, viscosity looses its meaning in the presence of strong translational symmetry breaking in the UV.

Stressing the relationship with dissipation, in Ref. [59] Hartnoll et al. find a loophole: the $\eta/s$ ratio acquires in a relativistic theory a more general meaning which makes sense also in the absence of momentum conservation. They show that this ratio is as well related to entropy

production via

$$\frac{\eta}{sT}\frac{d\log(s)}{dt} = 1. \tag{11}$$

For the minimal viscosity $\eta/sT \simeq \tau_\hbar$: the ratio of shear viscosity to entropy density is associated with the time scale setting the increase of the logarithm of the entropy density, becoming the Planckian time dealing with the minimal viscosity!

Hartnoll et al. use this subsequently to compute numerically the viscosity for the various linear axion- and Q lattice cases [59]. They rely on the standard Kubo formula result

$$\eta = \lim_{\omega\to 0}\frac{1}{\omega}\mathrm{Im}\,G^R_{T^{xy}T^{xy}}(\omega, k = 0), \tag{12}$$

where $G^R$ is the retarded propagator associated with the (spatial) shear parts of the energy-momentum tensor.

A finding of potential significance in the present context is that in the cases where the potential is irrelevant in the IR it disappears actually rapidly, with the implication that there is a large temperature range governed by emergent Galilean invariance and hydrodynamic behaviour. The $\eta/s$ ratio using the $\eta$ computed by Eq. (12) becomes temperature independent in this low temperature regime.

But there seems to be an issue with the magnitude of the ratio in this low temperature regime. Define $\eta/s = A_\eta\,\hbar/(k_B T)$: the numerical factor $A_\eta$ turns out to be here typically much smaller than $1/(4\pi)$. This is at first sight quite confusing since it is generally believed that $1/(4\pi)$ should be a lower bound: the large N CFT is supposedly maximally entangled. This would actually also disqualify the proposal for the linear resistivity in the next section since we will see that this requires $A_\eta > 1/(4\pi)$.

This conundrum was resolved in Ref. [60]. In fact, the shear-shear propagator of a hydrodynamical system is

$$G^R_{T^{xy}T^{xy}}(\omega, k) = \frac{\omega^2\eta'}{i\omega + \mathcal{D}k^2}, \tag{13}$$

where the diffusivity $\mathcal{D} = \eta/(sT)$ in a conformal system: in the next section we will see that this $\eta$ is the one relevant to transport. However, it follows immediately from Eq. (13) that the viscosity measured by Eq. (12) is instead the $\eta'$ appearing in the numerator. This has actually the status of a spectral ("Drude") weight associated with the shear currents. In a Galilean invariant system $\eta = \eta'$ but this is not at all the case when the translational invariance is broken in the UV! The $\eta$ associated with the diffusivity is the IR quantity and explicit computation [60] shows that this takes yet again the minimal value $A_\eta = 1/(4\pi)$! The $\eta'$ is UV sensitive and it signals that the overlap between the UV degrees of freedom feeling the strong potential and the "Galilean" IR degrees of freedom is strongly reduced.

Turning back to the cuprates, the message of these holographic exercises may well be that the strong UV lattice potentials that are a prerequisite to form the strange metal are strongly irrelevant in densely entangled systems. This would have the desirable ramification that hydro behaviour will set in at quite high temperatures. With regard to the minimal viscosity, although it may be that the constancy of $\eta/s$ is a universal principle in such systems, much less is known regarding the dimensionless parameter $A_\eta$. The debate in the string theory community has been raging whether a lower bound exists: $A_\eta \geqslant 1/(4\pi)$. However, in the cuprates it may well be quite a bit larger than $1/(4\pi)$ given that the strange metals are presumably less densely entangled than the large N CFT's of holography. Such a "large" $A_\eta$ is actually desirable in the context of the ploy presented in the next section.

### 4.4.2 Finite density: is the minimal viscosity a large N artefact?

The minimal viscosity is firmly established in the zero density CFT's. However, we wish to apply it in the context of the *finite density* strange metals. According to holography this does not make any difference for the minimal viscosity [61]. In the bulk it is associated with the universal nature of the absorption cross section of gravitons by the black hole. That the black holes are charged to encode for the finite temperature physics in the boundary does not make any difference in this regard.

However, holography should be used with caution. The trouble is with the large N limit required for the use of classical gravity in the bulk. A number of instances have been identified where this limit is responsible for pathological behaviors [7]. A well understood example is the mean-field nature of thermal phase transitions in holograpy: large N takes a role similar to large d in suppressing thermal fluctuations. Hartnoll and coworkers raised the alarm that the minimal viscosity in the charged systems may well be such a large N pathology [62]. This is controversial while it appears to be impossible to decide this issue on theoretical grounds. One may view the remaining sections as a proposal to use nature as analogue quantum computer to shed light on this foundational matter!

The arguments by Hartnoll *et al.* [62] rest on dimensional analysis. Let us consider again the dimensional analysis for the viscosity as I just discussed for zero density. Depart from $\eta = f \tau_q$; at finite density the free energy density becomes $f = \mu + sT$. When $\mu >> T$ the free energy density becomes $\mu$ instead of $sT$. Identifying $\tau_q$ with $\tau_\hbar$ it follows that $\eta \sim \hbar(\mu/T)$: the viscosity behaves as a "milder" version of the Fermi liquid one, $\sim (E_F/T)^2$.

Why worry? This dimensional analysis is after all rather ad hoc. There is however a more penetrating form of dimensional analysis relating it to natural properties of entangled matter [62]. Let us take the zero rest mass matter of the literal holographic fluid, having the implication that we should use the dimensions of relativistic hydro. It follows that one can relate the viscosity to the momentum diffusivity $\mathcal{D}$ using the energy density $\varepsilon$ and pressure $P$ via the relation $\eta = \mathcal{D}/(\varepsilon + P)$. For finite density and low temperature $\varepsilon + P \to \mu$ and the temperature dependence of $\eta$ should be the same as that of the diffusivity $\mathcal{D}$, with dimension $\dim [\mathcal{D}] = m^2/s$. Assert that $\tau_\hbar$ is the universal time associated with any relaxational process in the entangled fluid. This in turn implies that $\mathcal{D} \sim v_B^2 \tau_\hbar \sim v_B^2/T$. What should one take for the velocity $v_B$?

According to holography this "butterfly velocity" is an IR quantity that is temperature dependent, $v_B = T^{1-1/z}$, which is in turn consistent with $\eta \sim s \sim T^{(d-\theta)/z}$. However, Hartnoll et al. argue on physical grounds that this $v_B$ should be instead a UV quantity that cannot be temperature dependent. Diffusion means that one disturbs the system locally, and $\mathcal{D}$ governs the way that this disturbance spreads in space as function of time (the ubitquitous ink drop). Hartnoll et al argue that it should be tied to the "spreading of the operator": use the Heisenberg picture (time dependence is in the operators) and in every microscopic time step the operator grows bigger (e.g. $n_i \to n_i n_{i+1}$ by commuting $n_i$ with the Hamiltonian) and this "growth" of the operator is clearly governed by microscopic parameters. One sees immediately that such a constant $v_B$ has the ramification that $\eta \sim 1/T$, as in the case of the naive dimensional analysis.

## 5 The linear resistivity and the Planckian fluid

Hopefully sufficiently well equipped by the tutorial sections, you are now entering the business end of this text. Why should the momentum relaxation rate in the cuprate strange metal track closely the entropy? This is not a truly new story either but it is much easier to explain than a couple of years ago. It is actually *very* easy.

We discovered it in 2013 [9], playing around with a holographic set up that was quite

cutting edge back in the day. The point of departure is the conformal to AdS$_2$ metal; we liked it (and we still do) because it grabs two important properties of cuprate strange metals, the local quantum criticality and the Sommerfeld entropy (section 4.2). In order to address transport it is necessary to break translational invariance, since otherwise anything will behave like a perfect conductor given that its total momentum is conserved. Back then the cheapest way to encode momentum relaxation was in terms of so-called massive gravity. In the mean time there are many other ways to achieve this goal [8] but a don't worry theorem got also established: as long as one is dealing with "near momentum conservation" (section 2) it is all the same. This just means that the momentum relaxation rate (the width of the Drude peak) is small compared to the microscopic scales of the system like the chemical potential. This is typically satisfied in the cuprates. Combining these two ingredients, the holographic oracle produced for the temperature dependent resistivities perfectly straight lines, from zero temperature up to temperatures approaching the chemical potential [9]. As in experiment, the resistivity values varied from extremely good metal to very bad metal without any interruption of the perfect straightness of the line. In fact, even the residual resitivity vanishes at zero temperature (section 6). To the best of my knowledge this is up to the present day the only construction that impeccably reconstructs these most striking features of the linear resistivity.

In hindsight we found out how to reconstruct the mechanism behind this result, just resting on principles that I argued in the above may have a quite general status dealing with entangled quantum matter. These are equivalent to the following set of assumptions:

Assumption 1: The minimal viscosity principle (section 4.3) applies to the cuprate strange metal.

Assumption 2: The strange metal is local quantum critical characterized by a dynamical critical exponent $z = \infty$ (section 4.2).

Assumption 3: "Rapid hydrodynamization" is at work, in the specific sense that the electron system behaves like a hydrodynamical fluid despite the presence of quenched disorder on the microscopic scale (section 4.3).

The line of arguments leading to the linear resistivity is now very simple. We know from direct measurements that the optical conductivity follows quite precisely a Drude behaviour (section 2) at least at frequencies lower than temperature (see also section 6.1 ). The DC resistivity is therefore determined by

$$\rho_{DC} = 1/(\omega_p^2 \tau_K), \tag{14}$$

where $\omega_p$ is the plasma frequency and $\tau_K$ is the long wavelength momentum relaxation time. In fact, both the Drude weight $\sim \omega_p^2$ as well as the "transport" momentum relaxation rate $\tau_K$ may depend on doping, temperature and so forth. Since $T << \mu$ (the chemical potential) a strong temperature dependence of $\omega_p$ would be unnatural and it is not seen in experiment, nor in any holographic strange metal. The linearity in temperature is therefore entirely due to the temperature variation of the Drude width

$$\frac{1}{\tau_K} = B \frac{k_B T}{\hbar}, \tag{15}$$

where $B$ is a quantity of order one which may well be doping dependent (section 6.1).

Perhaps the most provocative claim follows from assumption 3: despite the translational symmetry breaking on the microscopic scale the electron liquid in the strange metals behaves as a hydrodynamical fluid. When this is the case, the origin of the resistivity is an elementary exercise in hydrodynamics. In order to dissipate the current/momentum in the Galilean continuum one needs spatial gradients in the flow, giving rise to a momentum relaxation rate $\mathcal{D}q^2$ where $\mathcal{D}$ is the momentum diffusivity of the liquid and $q$ the inverse gradient length. Upon breaking weakly the translational invariance one simply substitutes $q = 1/l_K$ in the $q \to 0$

limit, where $l_K$ is the length where the breaking of Galilean invariance becomes manifest to the fluid. I will call this in the remainder the "disorder length". When the Drude response is associated with a hydrodynamical fluid the transport momentum relaxation rate will therefore be

$$\frac{1}{\tau_K} = \frac{\mathcal{D}}{l_K^2}. \tag{16}$$

The temperature dependence of $\tau_K$ can be now due to both $\mathcal{D}$ and $l_K$. The latter quantity is just parametrizing the strength of the disorder potential and in a quantum critical liquid one expects it to be a coupling running under renormalization: $l_K$ is a-prori a scale- and thereby temperature dependent quantity. At this instance we have to invoke assumption 2. It follows from elementary scaling considerations that in a local quantum critical liquid *any* spatial property will not renormalize; the disorder strength is marginal and will therefore be scale-, and thereby temperature independent. All the temperature dependence has to come from the diffusivity.

The diffusivity is in turn proportional to the viscosity. I already introduced the relativistic relation between the two (section 4.4): $\mathcal{D} = \eta/(\varepsilon + p)$. In the non-relativistic electron fluid stress-energy is just completely dominated by the rest mass: $\varepsilon + p \rightarrow nm_e$, where $n$ and $m_e$ are the electron density and mass, respectively. It follows that

$$\frac{1}{\tau_K} = \frac{\eta}{l_K^2 nm_e}. \tag{17}$$

Up to this point we just needed elementary hydrodynamical wisdom that is used all the time by engineers, designing fuel pumps and so forth. But now we engage assumption 1, *the universality of the minimal viscosity*. Let us parametrize it as

$$k_B \frac{\eta}{s} = A_\eta \, \hbar, \tag{18}$$

where $s = nS$ is the entropy density and $S$ is the dimensionless entropy, while $A_\eta = 1/(4\pi)$ dealing with the large $N$ matter of holography. This may well take other values "of order unity", dealing with the UV conditions of the electron system. By combining Eq. (18) and Eq. (17) we have arrived at the *main result*:

$$\frac{1}{\tau_K} = \frac{A_\eta}{l_K^2 m_e} \hbar \, S, \tag{19}$$

demonstrating that *the resistivity has to be proportional to the (dimensionless) entropy $S$*. Notice that the density factors $n$ cancel; it is easy to check that the right hand side has the dimension of inverse time given that $A_\eta$ and $S$ are dimensionless. This is the exquisitely simple prediction as announced in the introduction that I would like to see thoroughly tested in the laboratory [63].

Before going into any detail regarding the test protocol, let us first find out whether it is violating any of the available experimental facts. According to Eq.(19) the Drude width only depends on the disorder length $l_K$ and $S$ asserting that there is just one strange metal in so far $A_\eta$ is involved. Surely $l_K$ may depend on doping but given the local quantum criticality the temperature dependence at fixed doping should be entirely due the entropy since $l_K$ is not running. In fact, since the early electronic specific heat measurements by Loram [64] it has been established that the only other quantity that is governed by a straight line as function of temperature is the entropy! This has the deceptively familiar Sommerfeld form

$$S = k_B \frac{T}{\mu}. \tag{20}$$

Subconciously this was perceived by many (including myself) as "the strange metal is somehow basically a Fermi liquid". However, as I discussed in section 4.2 this cannot be reconciled with the local quantum criticality, highlighting a holography inspired way out in the form of the conformal-to-AdS$_2$ scaling. Similarly, since it is not a Fermi liquid it is semantically improper to invoke the word Fermi energy. The chemical potential $\mu$ is the unbiased name for this quantity and it is measured directly from the magnitude of the Sommerfeld constant to be $\mu \simeq 1\,\text{eV}$ [64]. The conclusion is that the resistivity is linear in temperature, for the simple reason that it is proportional to the entropy that is directly measured to be linear in temperature as well. Can it be easier?

The next question is, do the absolute magnitudes of the various quantities make sense? Both the Drude width $1/\tau_K$ as the entropy have been directly measured, while $l_K$ is a-priori unknown. It should already be obvious that this length appears in transport quantities in a way that is very different from the usual Fermi-liquid lore. It is in way quite hidden dealing with the entangled quantum matter. However, we can now profit from the fact that it collapses at least to a degree in the underdoped regime into classical stuff: the charge order. It is well documented that this charge order is quite disorderly on large length scales due to the interaction with the quenched disorder [66]. The characteristic disorder length may well be a good proxy for $l_K$ and these data suggest it to be of order of 10 nm. Let's see whether this fits the linear resisitivity numbers. Inserting the measured entropy Eq. (20) in Eq. (19),

$$\frac{1}{\tau_K} = \frac{A_\eta \hbar}{l_K^2 m_e} \frac{k_B T}{\mu} \tag{21}$$

observe that this can be written as,

$$\frac{1}{\tau_K} = A_\eta \frac{l_\mu^2}{l_K^2} \frac{1}{\tau_\hbar}. \tag{22}$$

We recognise the familiar Planckian dissipation time $\tau_\hbar = \frac{\hbar}{k_B T}$, but in addition a "chemical potential length"

$$l_\mu = \frac{\hbar}{\sqrt{m_e \mu}} \tag{23}$$

can be defined balancing the disorder length scale $l_K$. This has a similar status as $1/k_F$ in a Fermi liquid, and filling in $\mu \simeq 1\text{eV}$, $l_\mu = \hbar/\sqrt{\mu m_e} \simeq 2$ nm. We can use now Eq. (22) for an order of magnitude estimate of $l_K$. According to the optical conductivity measurements $\tau_K \simeq \tau_\hbar$ and assuming $A_\eta \simeq 1$ it follows directly that $l_K \simeq l_\mu$, in the nanometer range! In fact, it is a bit disqueting since $l_K \simeq l_\mu$ implies that the disorder scale hits the UV scale $\mu$ and under this condition assumption 3 (quantum thermalization outrunning the disorder) gets seriously challenged. This is however just dimensional analysis and in reality there may well be parametric factors changing this estimate considerably.

Let us now judge this affair on qualitative grounds. It actually seems to resolve the mystery associated with the linear resistivity in a most natural way. How can it be that the resistivity stays strictly linear, from low temperatures where the strange metals are very good metals up to high temperatures where it turns into a very bad metal? Taking the estimate of the disorder length to be in the nanometer range would imply that in case it would form a conventional Fermi liquid it would already be at zero temperature a quite bad metal characterized by a large residual resitivity. However, dealing with a hydrodynamical liquid the resistivity is associated with the viscosity. Assuming the minimal viscosity to be at work, the magnitude of the resistivity is determined by the entropy. The entropy is in turn like the one of a Fermi gas, and it follows from direct measurements that the entropy becomes very small at low temperatures:

holographic strange metals mimic in this regard the Fermi-gas notion that at low temperatures the microscopic degrees of freedom have disappeared in the Fermi sea. The consequence is that the viscosity becomes very small at low temperatures; despite the presence of quite sizable disorder the fluid stays very conducting because of its very small viscosity. Upon rasing the temperature the viscosity increases with the entropy and in the bad metal regime it starts to become of order of the viscosity of normal fluids which would show the same kind of nominal resistivities as observed in cuprates at high temperature. Surely, the ingredients of the Ioffe-Regel limit are just not in existence [21]: there are no particles having a mean free path that can be compared to lattice constants.

Another striking feature is that it offers an appealing explanation for another long standing mystery: at least in good crystals and thin layers lacking grain boundaries and so forth the residual resistivities are remarkably small. Stronger, in the best samples at optimal doping it appears that upon extrapolating the normal state resistivities to zero temperature there is no intercept, the residual resistivity disappears completely! [67]. This was recently confirmed even in the "dirty" 214 system, using extremely high quality MBE grown thin layer samples [68]. The explanation in the present frame work is very easy: upon approaching zero temperature the entropy is vanishing and thereby the viscosity. The prediction is that when the metal could be stabilized in zero magnetic field and zero temperature a *perfect* fluid would be realized. Although such a fluid is not a superconductor and magnetic fields can penetrate, it would nevertheless be a perfect conductor! I will take up this theme in the second part of next section.

# 6 The predictions

One discerns a simple principle that has strong predictive power – the Drude width should scale with entropy – offering perhaps the first truly credible explanation of the linear resistivity. However, if correct it would prove a radically new form of physics to be at work: universal properties of densely entangled matter shared with for instance the quark gluon plasma. Given the gravity of the claim, *overwhelming* experimental evidence is required.

There is indeed a lot more predictive power than just in the a-posteriori explanation of the linear-in-T resistivity. The first set of predictions revolve around the *doping dependence of the linear resistivity* (section 6.1) which just needs routine research, be it quite some work. The second set of predicitons revolving around *nanoscale turbulence* do require the development of novel experimental machinery which may be in practice not easy to realize. If these predictions would be eventually confirmed they will represent an undeniable, unambiguous evidence proving once and forever that the basic assumptions listed in the introduction of section 5 are indeed at work.

## 6.1 The doping dependence of the resistivity

As announced in the introduction, the writing of this paper was triggered by the observations of Legros *et al.* [2]. These authors claim that the dependence on doping of the pre-factor of the linear DC resitivity tracks closely the doping dependence of the measured Sommerfeld constant, from optimal doping up to rather strongly overdoped. Being aware of Eq. (19) this of course rings a bell. These authors actually forward another explanation. The DC resistivity is set by Drude weight and Drude width; they claim the latter to be mere Planckian $\sim 1/\tau_\hbar$ arguing that the doping dependence is associated with the Drude weight. This would be quite reasonable in a Fermi gas where the Drude weight and the thermodynamic density of states are the same thing in two dimensions. This is however extremely special for the non-interacting Fermi gas: fundamentally there is no reason that these quantities should be related in a direct

way. This was already highlighted in the early 1990's in the cuprates in the form of the theme of spectral weight transfers in doped Mott insulators. Upon doping such an insulator the low energy optical spectral weight increases much faster in the cuprates than the nominal doping density [69], for many body reasons that are quite well understood [70]. But there is even a deeper problem of principle with the interpretation of Legros *et al.*: the identification of Drude weight with the thermodynamic density of states is a free Fermi gas property, while the Planckian time requires in one or the other way many body entanglement. These two conditions are just mutually exclusive.

In this regard the *optical conductivity* is vastly superior to DC transport. From the frequency dependence one can infer whether it is in fact governed by a Drude form, and if so one can determine both the Drude weight- and width separately from the data. What needs to be done therefore is to measure on a single set of samples over the full metallic doping range in the temperature regime where strange metals occur: (a) the DC resisitivity, (b) the "Loram-Tallon" electronic specific heat (with an eye on entropy, see underneath) and (c) the optical conductivities. When the correlation between the entropy and the prefactor of the linear resistivity as claimed by Legros *et al.* [2] is confirmed, the optical conductivity should then reveal whether this correlation is due to the Drude width or Drude weight (or even both). Obviously, a strong correlation between entropy and Drude width would represent stunning evidence for the "minimal viscosity hypothesis". One uneasy aspect is that the overdoped regime may not be the best regime to look for linear resistivities, since there are good reasons to suspect that Fermi-liquid physics is too nearby [46]. However, the strange metal formed above the pseudo-gap temperature in the underdoped regime may be a more trustworthy terrain and here quite unexpected behaviors are predicted as I will explain underneath.

Before addressing this, let me first discuss some further caveats. Consider the functional form of the optical conductivity up to energies of order of an eV where interband transitions take over; this is well known to be not a straight Drude form. Also deep in the metallic doping regime long "tails" are found at higher energy and these have been fitted with a myriad of different fitting forms. For instance, one can keep the Drude form and employ a frequency dependent optical self-energy (the "generalized Drude"). However, asserting that the system forms a hydrodynamical fluid an extra constraint follows for the line shape. In the "hydrodynamical" regime of frequencies less than temperature the optical conductivity should have a *precise Drude form*: there is just one momentum relaxation time $\tau_K$ that should not depend itself on frequency. It is intrinsic to this physics that, given the rapid thermalization, the system behaves as a truly classical fluid characterized by this single time. In fact, high quality evidence for this being the case was already presented in ref. [22] for an optimally doped BISCO, in the form of a Drude scaling collapse of the low frequency data.

One would like to check this with more scrutiny. One approach is to parametrize the optical conductivity as $\sigma(\omega) = \sigma_{Drude}(\omega) + \sigma_{incoh}(\omega)$, to check whether the non-Drude high energy tail $\sigma_{incoh}(\omega)$ is indeed vanishing at low energy. Notice that this tail is identified with $\sigma_{incoh}(\omega) \sim \omega^{-2/3}$ branch-cut tail extending up to high energies [22]. This has some fame among holographists because of the claims found in ref. [73].

Very different from the underdoped regime, it was already observed in the mid 1990's that the total optical spectral weight obtained by integrating up $\sigma(\omega)$ to the cut-off becomes remarkably *doping independent* in the overdoped regime [71, 72]– this argues already against the assertion of ref. [2] that the doping dependence is in the spectral weight. This however deserves some caution: although the weight coming from the sum of $\sigma_{Drude}(\omega)$ and $\sigma_{incoh}$ is rather doping independent it can not be excluded that there is a substantial spectral weight transfer from the incoherent part to the Drude part as function of overdoping. To the best of my knowledge this has not been looked at in any quantitative detail. Obviously, the weight in the Drude part is the one of relevance to find out the correlation with entropy.

I am not aware of any published data where the Drude part of the optical conductivity has been systematically and quantitatively studied in the *underdoped* strange metal regime. The minimal viscosity hypothesis leads here to a rather unexpected prediction – if confirmed this would surely add substantial evidence to the case. According to Loram *et al.* [65] the strange metal realized in the strange metal regime at temperatures above the pseudogap temperature $T^*$ is characterized by a Sommerfeld constant that is within resolution the same as at optimal doping. One could be tempted to conclude that the momentum relaxation time should be doping independent for underdoped strange metals. However, there is a subtlety at work. Based on thermodynamic measurements, Loram and coworkers [65] arrived at the strong case that the pseudogap appears to be a temperature independent, rigid gap. It becomes unobservable at $T^*$ because this gap just fills up because of thermal distribution factors. Although the specific heat may look the same, the *entropy* becomes now doping dependent in the underdoped strange metal regime at high temperature. There is just an entropy deficit associated with the pseudogap taking the form of a negative off-set when extrapolated to zero temperature: $S = -S_0 + \gamma T$, which is not seen in the specific heat since $C = T(dS)/(dT)$. Given that the pseudogap is growing linearly with underdoping it is easy to find out that the entropy in this underdoped regime takes the effective form $S = (x_{opt} - x)\gamma T$ [74]. *It increases linearly with increasing doping in the underdoped regime!* Assuming that $1/\tau_K \sim S$ the Drude width should actually *decrease* with underdoping. It is well established that the prefactor of the linear resistivity is increasing for decreasing doping, e.g. [17] : it has been taken for granted in the past that the growth of $1/\tau_K$ should be the culprit. However, it is well documented that the Drude weight is rapidly decreasing when the doping is decreasing [69] because of the "Mottness" effects which may then explain the increase of the DC resitivity at a fixed temperature.

To end this subsection, a final warning concerns the disorder length $l_K$; a close correlation between entropy and Drude width as function of doping requires this quantity to be rather doping independent. I have not found out how to measure this quantity indepedently and it is surely very difficult to compute it with any confidence. At least in the overdoped regime one may wish to argue that the intrinsic Coulombic disorder due to the charged defects in the buffer layers should be the most obvious suspect. The strength of the effective disorder potential should then be governed by the screening properties of the metal. One may then argue that this becomes quite doping independent given the doping independence of the optical spectral weight. This becomes less obvious in the underdoped regime – there should surely be a point where $l_K$ should start to show doping dependence.

## 6.2 Minimal viscosity and turbulence on the nanoscale

Perhaps the most famous dimensionless parameter in physics is the Reynolds number. Two regimes are intrinsic to hydrodynamics. On the one hand, when dissipation dominates one is dealing with the *laminar* flow regime characterized by smooth flow patterns. On the other hand, under conditions where dissipation is small the non-linearities take over causing highly irregular *turbulent* flows. The Reynold's number is the dimensionless quantity that discriminates between the two regimes. From straighforward dimensional analysis of the Navier-Stokes equations it follows immediately that this number "*Re*" for a non-relativistic electron fluid is governed by

$$Re = \frac{\rho v_{tr} L_{tr}}{\eta}, \tag{24}$$

where $\rho = n m_e$ is the electron mass density ($kg/m^3$), $v_{tr}$ the characteristic transport velocity ($m/s$), $L_{tr}$ the characteristic linear dimension of the flow($m$) and $\eta$ the (dynamical) viscosity ($kg/(ms)$). The viscosity embodies the dissipative power of the fluid, and its dimension is balanced by the quantities in the numerator. As a rule of thumb, when *Re* is order unity the

flow wil be laminar under all conditions. For $Re > 10^3$ the system will be deep in the turbulent regime. For $Re \simeq 50$ one encounters preturbulence phenomena such as the Karman sheets: vortex patterns that emerge at cylindrical obstacles put in the flow [75] .

One observes that the linear dimension of the flow $L_{tr}$ appears in the numerator of Eq. (24). The consequence is that for standard fluids (like water) it is impossible to find turbulent behavior on scales less than roughly a millimeter. It is for instance overly well established that in the micron regime of microfluidics and biology the flows are invariably deep in the laminar regime; bacteria do swim in a very different way from humans. But this is of course tied to the typical value of the viscosity $\eta$.

How does this work dealing with the minimal viscosity? It is actually a frontier of fluid-gravity (holographic) duality. Pushing the frontiers of the numerical GR in the bulk, Schessler et al. showed that there is a turbulent regime in the boundary fluid, being dual to black hole horizons with a *fractal* geometry [76] ! However, this does not play a role in the context of the quark gluon plasma: although $\eta/s$ is minimal the entropy density is large under the condition at the heavy ion colliders with the effect that the absolute value of the viscosity is actually quite large.

However, an interesting surprise happens relying on the assumption that the minimal viscosity survives the hyperscaling violation (section 4.4). Dealing with a Sommerfeld like entropy $s = nT/\mu$ the entropy becomes very small at low temperature as is actually directly measured in the cuprate strange metals [64]. The ramification is that the viscosity becomes extremely small, in turn implying that the fluid should be extraordinarily susceptible to turbulent flow behavior. This can be easily quantified. For the Reynolds number only the viscosity matters as fluid parameter. Inserting the result $\eta = A_\eta \hbar n k_B T / \mu$ and the definition of $l_\mu$, Eq. (23),

$$Re = v_{tr} \, L_{tr} \, \frac{1}{A_\eta} \, \frac{\tau_\hbar}{l_\mu^2}. \tag{25}$$

To get a better view on the dimensions, it is helpful to identify an intrinsic velocity. In the Galilean continuum the holographic strange metals are characterized by a zero sound mode (zero temperature compression mode) with a velocity set by the chemical potential [61]. For the non-relativistic electron system this should correspond on dimensional grounds with a velocity that is similar to the Fermi velocity of a Fermi liquid (identify $\mu \to E_F$): $v_\mu = \sqrt{2\mu/me}$. By using EELS one can measure the charge density dynamical susceptibility directly and this was already accomplished by Nuecker et al. in the early 1990's [77] and very recently reproduced at a much higher energy resolution by Abbamonte *et al.* [44] This shows directly the existence of a *plasmon*. The plasmon is clearly dispersing with a characteristic velocity $v$ which is of the same order ($\simeq 1$ eV A.) as $v_\mu$.

In terms of these units, the expression for the Reynolds number becomes remarkably elegant,

$$Re = \frac{\sqrt{2}}{A_\eta} \frac{v_{tr}}{v_\mu} \frac{L_{tr}}{l_\mu} \frac{\mu}{k_B T}, \tag{26}$$

expressing the length ($L_{tr}$) and velocity ($v_{tr}$) dimensions of the hydrodynamical flow in the natural units of length $l_\mu \simeq 1$ nm and velocity $v_\mu \simeq 1$ eV A, the Reynolds number is of order $\mu/(k_B T)$!

To assess how these dimensions work we need one more characteristic dimension. Our system is spatially disordered and beyond a charactistic length $\lambda_K$ momentum is no longer conserved and it will be impossible to generate turbulent flows, which will be completely dissipated by the "obstacles in the flow". In a free system this would be coincident with the microscopic disorder length but this is not the case in the hydrodynamical fluid. It is instead

governed by the viscosity: in the perfect fluid ($\eta = 0$) this length would of course diverge. $\lambda_K$ is very easy to estimate: it is the length scale associated with the momentum relaxation time $\tau_K \simeq \tau_\hbar$ and we just identified $\nu_\mu$ being the quantity with the dimension of velocity

$$\lambda_K = \nu_\mu \tau_K. \tag{27}$$

This quantity will play the role of an upper bound for the length $L$ in the Reynolds number: flows on a scale larger than $\lambda_K$ have to be smooth.

We have now estimates for all the numbers that matter. The chemical potential is $\mu \simeq 1$ eV $= 10^4$ K. Let us consider an optimally doped cuprate with a "low" $T_c = 10$K. It follows that $\lambda_K \simeq 100$ nanometer, shrinking to 10 nm at 100K. In order to stand a chance to observe the turbulence one has to construct nano-transport devices of the kind that have been realized with much succes in e.g. graphene [78] (see also [79]). This is the burden for the experimentalists because it has proven to be very hard to construct such devices involving cuprates with their hard to control material properties.

I started arguing that turbulence should not occur even on the micron scale, let alone on the nano scale. But now we find the dimensions of the minimal viscosity fluid on our side. The big deal is the ratio $\mu/(k_B T) \simeq 10^3$ at 10 K: this sets the intrinsic magnitude of the Reynolds number according to Eq. (26)! Let us take $ł_\mu \simeq 2$ nm as before and consider a typical device dimension of 20 nm. It follows that $Re = 10^4 (v_{tr}/v_\mu)$: when a "high subsonic" transport velocity $v_{tr} \simeq v_\mu$ could be taylored $Re$ would be 10,000 and the system would be very deep in the turbulent regime!

This represents the next great challenge for the device builders. One imagines a device of the kind as realized in graphene [78] where one injects a current jet through a narrow constriction, looking at the current patterns through nearby "probe" constrictions. It may again be for practical reasons (heating, etcetera) very difficult to realize very high transport velocities. Yet again, the $\mu/k_B T$ "dimensional elephant" is of much help: even when the transport velocity is a mere percent of $v_\mu$ $Re$ is still of order 100. In this regime one already expects the Karman flow like preturbulence phenomena. These have already been explored to a degree in the context of graphene at charge neutrality [80] where one meets a similar overall situation [81].

The big picture message should be by now loud and clear. According to the dogma's of particle physics it should be impossible for electron flows to behave hydrodynamically in a system as dirty as the cuprates - one needs an exquisitely clean system such as the best quality graphene. Even when such a hydrodynamical fluid is realized, the colliding particles will give rise to a viscosity which is by order of magnitude of the same kind as found in daily life fluids like water – this is actually the case with the Fermi-liquid in finite density graphene [78]. Such viscosities are too large by six orders of magnitudes or so to allow for any form of turbulent flow phenomena on the nanometer scale. There would be no room for any doubt left when eventually such nanoscale turbulence would be observed in cuprates. Such a dramatic observation would require an explanation in terms of truly new physics. I hope that I have convinced the reader in the mean time that this is no longer an incomprehensible mystery: the nanoscale turbulence is on quite general grounds just to be expected dealing with genuine, densely entangled compressible quantum matter.

Let me finish this story by putting a well known, but puzzling experimental observation in the limelight, in a last attempt to lure experimentalists to take up this very risky endeavor. I already highlighted the old puzzle of the small, and even vanishing residual resistivities especially near optimal doping [67, 68]. Surely, this cannot be explained of particle physics. But how does this work departing from the minimal viscosity? It is just a natural condition for ground states to be non degenerate: ground state entropy is pathological. This also applies to holographic strange metals: initially there was some confusion since the simplest RN metal has ground state entropy but this turns out to be a singular case [7]. Now it gets very easy:

the viscosity is proportional to entropy and when the entropy is vanishing at zero temperature the viscosity is vanishing. A fluid with zero viscosity is a perfect fluid and a perfect fluid flow is unstoppable! The conclusion is that the minimal viscosity hypothesis *implies* a vanishing of the residual resistivity.

To the best of my knowledge there is at present no other explanation for the absence of residual resistivity. Although I have no clue how to prove it, there is a way to argue that it may well be a general property of ground states of many body entangled metalic states of matter. It is helpful to exploit an analogy with regular superconductors. Consider a system of hard core bosons in first quantized, position space path integral language. It is well understood how this works: below the Bose condensation temperature worldlines of bosons emerge that wind an infinite number of times along the imaginary time cylinder [31, 82]: this is just the path integral way to recognize that in real space every boson is entangled with every other boson and this is quite like many-body entanglement. However, upon dualizing to momentum space/phase representation one finds that it is actually a product state that in turn breaks $U(1)$ giving rise to the Meissner effect, etcetera. To understand the vanishing of the resistivity one may as well refer to the real space representation: the "tangle of worldlines" turns the system of bosons in an infinitely slippery affair that cannot be brought to a standstill by anything in real space. It may well be that the many body entangled metallic state is in this regard similar to the superconductor, with the difference that no representation exists where it turns into a SRE product, breaking a symmetry.

It would be wonderful when such a state could be identified at virtually zero temperature. The cuprates fall short since superconductivity intervenes. Upon suppressing the superconductivity with a large magnetic field, the residual resistivity is rising linearly in the field [68] which is not understood even in a holographic language. In addition, one finds indications of the resurrection of particle physics in the form of the quantum oscillations that are naturally explained in terms of the Fermi-liquid [83]. It may well turn out to be that a many body entangled ground state is infinitely fragile – it is after all much like the states that have to be kept stable in the innards of the quantum mainframe computers that may be built in some future. However, temperature may come to help in the sense that the finite temperature properties become of the kind associated with the entangled ground state. Observing the nano turbulence would represent firm evidence for at the least this to be the case.

# 7  Epilogue

Establishing by precision experiments that the minimal viscosity is indeed at work in the cuprate strange metals would be undoubtedly a great stride forwards in establishing that the physics of compressible quantum matter is at work. Would it answer all questions? In fact, there are quite a number of issues that I swepted deliberately under the rug. I focussed the attention on the "good" strange metal realized at "low" temperatures in the cuprates because this is the regime where the minimal viscosity interpretation is most straightforward.

However, what happens upon raising the temperature to values deep into the "bad" strange metal regime? As highlighted in ref. [84], at temperatures well above the Ioffe-Regel limit it becomes questionable whether one can get still away with the "near-hydrodynamical" assumption 3 of section 5. One finds that the optical conductivity is no longer of a Drude form. The zero frequency conductivity becomes smaller and smaller with raising temperature and one finds that instead a Lorentz oscillator peak develops at finite frequency with an energy and width that both – mysteriously – increase linearly in temperature [84]. Because of the damping the weight of this oscillator is spilling over all the way to zero frequency, becoming responsible for the linear in T DC resistivity at high temperature. One way to interpret this is in terms

of the pinning mode of a fluctuating charge order [85]; perhaps not an entirely crazy idea given the evidences from (unbiased) quantum Monte Carlo simulations [86] that already at very high temperatures spin-stripe correlations are building up. However, one anticipates that such pinning modes should come down in energy for increasing temperature since the order is decreasing: this is even true in explicit holographic constructions of pinned charge order [42]. The real trouble is with the Laughlin criterium: how can it be that a qualitative change in conduction mechanism from perhaps a minimal viscosity type at low temperatures to a fluctuating charge order at high temperatures does not interrupt the linear resistivity? Very recent holographic modelling may shed an intriguing "unparticle physics" light on this puzzle [87]. Departing from the conformal-to-AdS2 holographic metal a "linear axion" is added to the bulk mimicking the fluctuation density wave. The outcome for the optical conductivity is the minimal viscosity affair of the above at low temperature while at high temperature it turns into the fluctuating charge order response. However, the claim is that graviational horizon universality takes care that the DC resistivity continues to be a straight line [88].

A difficulty which is closely related is the role of the phonons. In Fermi-liquids phonons are *the* source of the electron momentum dissipation at elevated temperatures. At present it is unclear how electron-phonon coupling works in holographic quantum matter for the reason that it is technically challenging to account for it in the graviational dual – it is still under construction [89]. However, very recent photoemission work seems to indicate that the electron-phonon is actually strongly enhanced in the strange metal regime [90]. In fact, the phonon thermal conductivity at high temperatures turns out to be quite anomalous indicative of a very strong coupling to the strange metal electrons to the degree that the thermal diffusivity is determined by the Planckian time [91]. These experimental results are a challenge for the assumption 2 in section 5: it is far from clear whether the local quantum criticality suffices as a mechanism to render the disorder length $l_K$ to be temperature independent when electron-phonon coupling is part of the equation.

Last but not least, there is more to Planckian dissipation than only the cuprate strange metals. The observation by Bruin *et al.* [16] that the Planckian time $\tau_\hbar$ can be discerned in the transport of even conventional metals in their high temperature bad metal regime is quite shocking. This has surely nothing to do with the minimal viscosity. Momentum conservation is irrelevant in these truly bad metals. Instead, the Planckian time scale may well enter through the charge- and energy diffusivities as conjectured by Hartnoll [21]. Undoubtedly, this is yet again rooted in quantum thermalization but one of a very different kind where electron-phonon coupling is likely playing a central role.

Linear resistivities are also observed in other systems than the cuprate strange metals. These show up in a quite restricted subset of heavy fermion systems and pnictides, while the linearity only extends over a limited temperature interval. Different from the cuprates, all these cases are associated with clear cut incarnations of isolated quantum critical points where a manifest order parameter seizes to exist at zero temperature. So much seems clear that the minimal viscosity mechanism does not apply. It is just natural for such conventional quantum criticality that the specific heat diverges right at the quantum critical point (QCP) and this is also what is observed. The resistivity is linear in the temperature regime where the specific heat rapidly increases, excluding a simple linear relation between these two quantities. Adding to this confusion is the very recent observation that I already announced [68]: at very low temperatures and very high fields one can swap the field energy for temperature in the Planckian time setting the resistivity in cuprates. It is still waiting for a *simple* explanation of this scaling behaviour – it has actually been identified in a rather involved holographic set up [92]. In fact, this scaling behaviour was first observed in a quantum critical iron superconductor [93] suggesting that at least in this regard these systems are ruled by common principle.

## Acknowledgements

The list of physicists that have been helpful in getting this story straight is just too long to mention here. However, a couple of you do deserve a special mention: Koenraad Schalm and Richard Davison as my partners in crime. Erik van Heumen as sparring partner in discerning the experimental strategies. Blaise Gouteraux for pressing the covariant-to-scale transformation in my head. Sean Hartnoll for his transport teachings in general, and specifically for the subject matter explained in section IV D. Obviously, Subir Sachdev who has managed to keep me infected with the Planckian bug since the early 1990's. The origin of this text is in the form of an informal note that played its role in a vigorous but insightful debate with Andre Geim. Last but not least, I acknowledge moral support by David Gross and Kip Thorne, both insisting that pursuits like this require perseverance rooted in responsibility to the cause of physics.

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
