# Peer review of "Planckian dissipation, minimal viscosity and the transport in cuprate strange metals"

_SciPost Physics, doi:SciPost Phys. 6, 061 (2019)_

## Round 3 · Referee Report · Anonymous (Referee 1) · 2018-11-26

Strengths

see report

Weaknesses

see report

Report

My overall view of this paper is that it could be published. I do not agree with all of the paper, and I will explain why below, but I think many of these disagreements are at a 'high level' -- namely, this paper presents a broad perspective on a problem that should be published. I would encourage the author to make some revisions at least acknowledging these concerns. Ultimately, however, if this 'provocative' paper were to focus experimental attention on some of the important questions described here, my concerns above will be minor in the end.

1) The logic of Section 2 to me makes little sense. Regardless of simplicity or not, quasiparticle based transport theory of electron-phonon scattering is well verified in many experiments. It is also a good example of how there can be lots of dimensionless parameters in a metal and in principle this is also true at strong coupling: e.g. T/E_F, or T/E_bandwidth.

2) One might worry that because holographic duality gives a solution to certain models via classical gravity, such models are necessarily not NP hard, so why will such models have anything to do with non-Fermi liquids if non-Fermi liquids are NP hard? I understand the logic of Section 3.1 but think it is a little bit loose and encourage the author to tighten up the last few paragraphs.

3) "Unparticle" refers to something specific posited by Georgi and I am unsure it's relevant to the discussion in Section 3.2.

4) Section 3.3 seems more philosophy than physics. Also, I don't understand the statement that at weak coupling ETH reduces to classical physics (kinetic theory?) and feel this needs some more exposition in the text. Finally I don't think the Planckian time scale has in any formal way been recast in ETH language, if this is even possible.

5) I liked the discussion at the bottom of page 18. But I think \theta = -\infty would only make sense in a large N limit: see e.g. arXiv:1504.02478. Perhaps the z->\infty is not so conclusively proven...

6) Holographic transport is a very messy field, I don't agree with the last sentence of the first paragraph of Section 4.3 that there is any universality at play. One can get very complicated transport physics in the bulk by adding dilatons etc, so I don't see very many "general ramifications" at play.

7) v_B is not the same as the Lieb-Robinson velocity in general. I don't think the dimensional analysis that v_B is T-independent is compelling (it may be the case but in the deep IR there is no profound reason this needs to remain).

8) It is not clear to me what viscosity means in a disordered system without momentum conservation, and in particular why the 'minimal viscosity' principle should keep holding. I think Assumptions 1 and 3 in Section 5 are very strong.

9) Is Eq. 20 compatible with a sensible requirement that l_mu >= v_B h/kT? I am not sure if l_mu = 2 nm... beyond some possibly pathological holographic systems at z=\infty I am not sure that hydrodynamics would make sense below this 1/T-dependent scale. If this hydrodynamic argument is not obeyed I see no reason for the simple momentum diffusion argument on the previous page to go through.

10) Why is it mysterious that smaller residual resistivities are associated with cleaner crystals (top of page 28)?

11) I am not sure I agree with the logic on page 29, there are plenty of examples of theories (including N=4 SYM) where thermodynamic coefficients are not strongly sensitive to weak vs. strong coupling. It is believable to me that weak coupling thermodynamics is a good approximation to the cuprate strange metals. What the authors of Ref. [2] did is not inconsistent as far as I can tell.

12) I am skeptical about looking for electronic turbulence in a strongly momentum relaxing environment.

13) I thought it was interesting that the author found that the universality of T-linear resistivity found by Bruin et al would not be related to a universal T-linear mechanism whereas the cuprate materials alone would exhibit a universal relation. I think some further comments here might be useful. For example what is special about the cuprates (proposed quantum critical phase vs. point?) that makes this hold?

Requested changes

see report

  • validity: good
  • significance: good
  • originality: ok
  • clarity: ok
  • formatting: good
  • grammar: good

Author:  Jan Zaanen  on 2019-01-22  [id 408]

(in reply to Report 1 on 2018-11-26)

This referee report signals that this tutorial is already exerting its beneficial influences. As I stressed in the submission letter, its originality is mostly in the claim that I may have found a way to tell a quite new story shared by a theorist's community in very simple words. It is explicitly aiming at informing the experimental community but there is also a hidden agenda. The string community at large, and especially the condensed matter oriented holographists have been in the rear guard with regards to assimilating insights from quantum information. These have the beneficial effect of demystifying to quite a degree what AdS/CMT is about. This is the central theme of my tutorial. The Eigenstate thermalization idea (the working horse of my text) appeared for the first time only some 2 years ago, with the effect that it has not yet disseminated in the whole community. I know from direct experience that at Stanford, MIT and Harvard it is now standard knowledge. On the other hand, I attended very recently a meeting of the European AdS/CMT community where I was the only person using this language in a talk. This referee is clearly not yet enlightened as is obvious from his critique -- see underneath. The ideas are quite simple, but it seems that any mind needs time to get used to it. I expect that very soon also this community will have embraced these insights, and my tutorial may help.

The referee is showing good taste expressing that these "high level disagreements" should never play their part in publishing decisions, and he presents a number of issues that I should consider in order to improve the text. Unfortunately, without exception the referee is missing the point and I cannot discern any instance where I see a need to change anything in the text.

Let me list in detail my rebuttal to the issues he is raising:

  1. "Logic of section 2": perhaps the referee should read section 2 again $\textit{[edited by moderator]}$. I just report here on a community consensus that developed in the first 20 years of high Tc research, formulated in a particularly lucid way by Laughlin. I am surprised that this referee is unaware since this is understood by many holographists.

  2. The first instance where the referee is missing the key insight. Holography computes by construction the "expectation values of local operators", the "output information" where the gross moral of the story is that these can be "unreasonably simple". In terms of complexity the bulk GR may therefore be extremely simple. I prefer to go slow in section 3.1, given that later on it appears that even this referee is missing part of it.

  3. "Unparticle physics": I remember well receiving an e-mail by Sachdev shortly after Georgi posted the paper, stating that finally also the high energy phenomenologists did see a light. As it turns out, Georgi came up with a very contrived way to derive CFT propagators $\textit{[edited by moderator]}$. But he invented the "unparticle" word that became increasingly popular in the present context. There is a need for appropriate semantics and I like it a lot more than the vague and imprecise "incoherent continua". It is already commonly used -- it gives away that this referee is not in the loop.

  4. This is perhaps the most painful comment by the referee: section 3.3 contains an elementary introduction to the notion of Eigenstate Thermalization. Calling this philosophy just signals that the referee has not at all understood what it is about. No wonder that later on he is just not capturing the story line. Apparently he did not take the effort to study ref's 13 and 37 where in elementary ways it is explained how the "classical analogue" system arises (indeed, kinetic gas theory but even chemistry!). In fact, in a strict mathematical sense Planckian dissipation is derived as an ETH phenomenon already in the central chapter of Sachdev's "quantum phase transition" book dealing with the transversal field Ising model. One may even quote AdS/CFT itself in this regard showing Planckian dissipation through e.g. the minimal viscosity while it is firmly established that large central charge CFT's are "maximally" entangled while AdS/CFT computes VEV's of local operators. I do not delve deeply in this issue since the text is in first instance serving experimentalists.

  5. I find this also uneasy but I am of the strong opinion that in the absence of any independent knowledge regarding the finite density boundary one better distrusts any interference from the bulk alluding to large N artefacts. It is a sad fact that nobody knows how to write the finite N quantum gravity theory.

  6. In the last paragraph of this section I emphasize the messy nature of holographic transport theories. In the first paragraph I do not claim universality but instead that new principles are at work. Conventional "particle physics" surely submits to general principle, it can be yet very messy. One more instance where the referee does not capture what is written.

  7. Here the referee and the author are on the same page. However, section 4.4 is dedicated to the interesting objections by Sean Hartnoll that I just reproduce here without own contributions.

  8. Assumptions 1 and 3 are indeed the key, perhaps landing somebody (not me) a Nobel prize if confirmed by experiment since these defeat any conventional intuition. A strange myth has been spreading in the AdS/CMT community that cuprate strange metals are very bad/disordered. It is however experimental fact that at low temperatures these turn into very good metals characterized by sharp Drude peaks. When hydro is realized (assumption 1, requiring ETH/dense entanglement) this implies that these are in the "nearly hydrodynamical regime" where one can surely identify a viscosity. According to holography minimal viscosity is universal since it is set by the area of the black hole horizon which is always present at finite temperature. Whether holography can be trusted in this regard is the subject of section 4.4.

  9. According to holography l_\mu >= v_B h / k_B T is not sensible, see section 4.4. One should realize that Hartnoll's use of v_B is no more than dimensional analysis. In this passage of the paper you find out another way that the dimensions can be tied together. Once again, I depart from real holographic solutions and I trust that more than intuitive dimensional analysis. However, at the end of the day, only quantum computers or experiment can decide it.

  10. Nothing of the kind can be found in this paragraph: the referee should read what is written here.

  11. The referee should know better: he refers to zero density CFT's where the power of conformal invariance "equalizes" thermodynamics. But this is surely not at all the case in finite density systems: case in point is the thermodynamics associated with EMD scaling geometries that are unrecognizable departing from weakly interacting systems. The authors of ref. 2 are completely unaware of this. Furthermore, it is surely not the case that in strongly interacting systems there is any relation between Drude weight and thermodynamic density states. This is their assumption which is strictly tied to the free (Fermi-liquid) fixed point. There are actually plenty of examples of condensed matter computable systems that violate such an equality.

  12. I am sceptical myself. However, it is a prediction for experiment with smoking gun status, that would land somebody (not me) a Nobel prize if confirmed.

  13. I kept this silent on purpose. I am not convinced that the claims by Bruin et al are correct. There is quite some cherry picking in the experimental community, such as that linear T resistivities are only seen over a small temperature range. I perceive the acoustic-phonon-linear-in-T seen in simple metals as a mere coincidence. The reason for the focus on cuprates is explained in section 2. I actually share this vision with some top experimentalists.

---

## Round 3 · Referee Report · Anonymous (Referee 2) · 2019-3-12

Strengths

See report

Weaknesses

See report

Report

This is an interesting and thought-provoking article, that I think deserves to be published ultimately. Prior to that, some arguments need to be tightened. In particular I believe the author should strengthen the theoretical arguments that motivated his proposal, including additional holographic calculations. He should also make more precise the discussion of the range of temperatures where he expects translation breaking effects to be negligible.

1) One central theme is that there is a universal relation between the shear viscosity and the electronic entropy density, and that this is supported by arguments coming from string theory. This statement is not completely correct. I do not wish to enter into the whole story of higher-derivative terms, although the author should keep this in mind. A different caveat to the universality of $\eta/s=1/4\pi$ is precisely the explicit breaking of translations. Going back to the author's own work [9], the resistivity is linear in $T$ at all temperatures, including when $T\ll \Gamma$, where $\Gamma$ is the momentum relaxation rate. This is because it is just proportional to the horizon area, or said otherwise the entropy density. This quantity is in turn completely determined by the scaling properties of the IR geometry $\theta\to-\infty$, $z\to+\infty$, $\theta/z=-1$.

However, in the presence of explicitly broken translations and even accepting as an operational definition of the shear viscosity the Kubo formula relating it to the low frequency limit of the imaginary part of the shear retarded Green's function, the shear viscosity computed holographically is not linear in $T$ all the way to $T=0$. As $T\lesssim \Gamma$, there is some subleading temperature dependence to $\eta/s\sim T^#$, which is actually governed by the IR scaling dimension of the shear component of the metric. Then $\eta/s$ would vanish as $T\to0$. This is explained in eg arXiv:1601.02757. This temperature dependence of the viscosity appears at odds with $\rho\sim T$ at the lowest temperatures, at least if the author insists that viscous effect are the dominant scattering mechanism at all temperatures of interest.

So what is needed here is an estimate of the temperature scale below which the assumption of approximate translation invariance breaks down. To be consistent with experiments, it should be lower than the temperatures at which the $T$-linearity of the resistivity has been experimentally reported. I think the author's estimate of the length scale $l_K$ should provide a rough value. What is it?

2) An alternative is that translations are restored at the IR fixed point. Then if this has $z=1$, $\eta/s\sim T^0$ is recovered, and the constant prefactor can be vanishingly small depending on the overlap between the IR emergent conserved momentum and the strongly dissipating UV momentum (see the discussion in section 5 of arXiv:1601.02757). Then the temperature dependence of the viscosity will once more be controlled by that of $s$. However I do not know if this can be compatible with $\rho_{dc}\sim T$. The author should check this more carefully, for instance starting from the solutions reported in 1601.02757, 1401.5436 or 1401.5077. Since translation invariance is restored in the IR, the ac conductivity would exhibit a sharp Drude peak as $T$ decreases.

3) Even if the concerns I raise in 1) can be assuaged, there remains the fact that the evidence the author presents comes from holographic models with a very specific type of translation breaking through massless scalars linear in the spatial coordinates. This is certainly not generic, and more general dissipation mechanisms do not relate the resistivity as straightforwardly to the entropy density. So my question to the author is the following: Tune your holographic system to a hyperscaling violating $z=\infty$ critical IR, so that the spatially averaged $s\sim T$ at low $T$; but now consider an inhomogeneous geometry where translations are broken by say random disorder. Under what circumstances will the momentum relaxation rate be governed by hydrodynamic operators? It is plausible to me that there will be regimes where non-hydrodynamic operators give non-universal contributions to the resistivity, which will have nothing to do with the viscosity and lead to totally different temperature dependence.

It could ultimately be that $\rho_{dc}\sim s$ is the right idea, but that this is unrelated to the viscosity, at least away from the bad metallic high temperature regime.

4) The discussion at the end of section 4.4 assumes that the diffusivity and the shear viscosity have the same temperature dependence $D\sim\eta\sim1/T$, and that this connects to the Planckian timescale $D\sim v_B^2 \tau_\hbar$ through the butterfly velocity. i) This velocity is not the same as the Lieb-Robinson velocity, as pointed out by the first referee. The Lieb-Robinson velocity is a UV, microscopic velocity that indeed determines the spreading of the operator with time in spin chains. The butterfly velocity is an IR velocity (in holography, a simple way to see this is that it is given by horizon data) that governs the spatial spread of the quantum chaotic growth of some out of time order commutator between two operators. ii) The butterfly velocity is not temperature independent, at least in holographic quantum critical phases with $z\neq1$, as can be easily checked from the existing literature. Rather, it scales like $v_B\sim T^{1-1/z}$, leading to a diffusivity $D\sim T^{1-2/z}$ if simple dimensional analysis applies. On the other hand, $\eta\sim s\sim T^{(d-\theta)/z}$. For $z=\infty$ fixed points, this gives $D\sim T$, $v_B^2\sim T^2$, $\eta\sim s\sim T$, which is consistent with $D\sim\eta\sim v_B^2 \tau_\hbar$. The temperature dependence of $v_B$ is crucial.

Requested changes

See report

---

## Round 4 · Author Response

This is ironic. This report is helpful, since I had overlooked the work by Sean Hartnoll et al, arXiv:1601.02757 but unfortunately the report of the second referee rests on an awkard miscomprehension of this work.

It looked at first sight detrimental for my story, albeit for other reasons than quoted by this referee. Write $\eta/s = A_\eta \hbar/k_B$. The suggestion from the Hartnoll paper is that in the presence of a strong UV potential $A_{\eta} << 1/(4\pi)$. In the dimensional analysis associated with the linear resistivity on page 28 this would have the consequence that the disorder length $l_K$ would become much smaller than the lattice constant - a fatal inconsistency. Fortunately Sean is next door and he pointed out immediately that the standard way to compute the viscosity holographically (Eq. 12 in the revised manuscript) is actually misleading in the present context. This was elucidated in a follow up paper (new ref. 62): as I explain in the revised version one has to inspect the full shear propagator to find out that the viscosity associated with the emergent hydrodynamical regime is the usual one with $A_\eta = 1/ 4\pi$! Surely, the referee missed this development and therefore his comments 1-3 are irrelevant anyhow.

Frankly, I am at a loss with regard to his train of thought in these remarks. As spelled out in Hartnoll's paper "their" $\eta$ has no a-priori relationship to transport: when for whatever reason Galilean invariance is broken in the deep IR there is no such thing as a hydrodynamical viscosity. More alarming, it seems that he did not pay any attention to the phenomenology of the cuprates as discussed at length in my paper. Dealing with nature itself one cannot pick at will from the repertoire of possible outcomes of a theory. One first looks at the constraints set by experimental information. There are independent empirical reasons to assert that $z \rightarrow \infty$. It is just fact that the optical conductivity of the strange metal reveals sharp Drude peaks at temperatures below 400 K or so. There is no experimental evidence whatsoever for relevancy of translational symmetry breaking (charge order) in the strange metal regime. We do know from experiment that the strong periodic lattice potentials disappear completely in the deep IR, while there are good reasons to assume that the quenched disorder is quite weak. Under these perturbative circumstances it is perfectly save to employ a minimal bulk set up such as massive gravity and the bottomline is that hydro behavior will come to an end at the length scale Eq. (27).

With regard to his comment 4: of course the butterfly velocity is temperature dependent according to finite density holography. Hartnoll's case is that he claims this to be unreasonable based on bounds for operator spreading. After much debate the two of us reached the conclusion that this issue cannot be decided on theoretical grounds alone, reason for me to include it in the "caveats" section. This comment just highlights that both referee's did not take the time to familiarize themselves with Hartnoll's reasoning.

---

## Round 4 · List of Changes

I am sceptical as well whether the minimal viscosity is at work. However, this is condensed matter physics and the role of the theorist is no more than to ask unusual questions to experiment, based on the capacity of equations to teach us to think differently. It may well be that the experimental answers are yet elsewhere but this represents usually progress. I added the original section 4.4 (related to the comment 4 of the referee) for the mere purpose to spell out to the experimental readership that holography is as any other theoretical physics activity in condensed matter physics to be looked at with distrust. But in my mind the "large N caveat" is even superseded by another one: it is very well understood that for the cuprate electrons to become interesting very large periodic lattice potentials are needed (Hubbard models, etc.). How can it then be that near perfect Galilean invariance physics emerges in the not so deep IR? I did not pay explicit attention to this obvious problem in the original paper for length reasons. I am grateful to the second referee for pointing out the Hartnoll viscosity paper (arXiv:1601.02757) because it represents a concise way to explain how it works in holography. Assuming that the linear axions have anything to do with the strong lattice potentials, the computations show that these are ${\em strongly}$ irrelevant. Upon departing from the UV, these dive down rapidly under renormalization, forming a quite large regime at low temperatures where \eta/s is constant signaling unambiguously the hydrodynamical regime. The lattice is more irrelevant than in e.g. a Fermi liquid where Umklapp switches on algebraically in the flow from the IR fixed point -- the $T^2$ resistivity.

For this reason I decided to expand the "caveat" section 4.4 in the revised manuscript. I added an overall introduction for rhetorical reasons. I text edited the "large N caveat" part (the original section 4.4), wiring in the temperature dependence issue of the butterfly velocity explicitly to help holographic readers like the present referee's to appreciate Hartnoll's point of view. This has now become subsection 4.4.2. I added a new subsection (4.4.1) that is relatively long for the reason that various aspects of the paper arXiv:1601.02757 (new reference 61) are quite informative in the present context. It also serves the purpose of advertising the insightful (new) ref. 62 that is apparently not yet widely disseminated in the holographic community (e.g., referee 2). I discuss the relation of $\eta/s$ to entropy production in the non-hydrodynamical setting, explain the presence of a large effectively Galilean invariant IR regime, to then elucidate the results of (new) ref. 62 showing that the emerging IR hydrodynamical viscosity is the minimal one. I conclude that a similar miracle should be at work in cuprates in order for the mechanism underlying Sections 5,6 to be possible altogether.

---

## Editorial Decision

published